# Projected Climatic and Hydrologic Changes to Lake Victoria Basin Rivers under Three RCP Emission Scenarios for 2015–2100 and Impacts on the Water Sector

**Lydia A. Olaka [1,2,3,]\***(image)**, Joseph O. Ogutu [4], Mohammed Y. Said [2] and Christopher Oludhe [2,5]**

1   Department of Geology, University of Nairobi, P.O Box 30197, Nairobi, Kenya
2   Institute for Climate Change and Adaptation, University of Nairobi, P.O Box 30197, Nairobi, Kenya
3   Stellenbosch Institute of Advanced Study (STIAS), Wallenberg Research Centre at Stellenbosch University, 7600 Stellenbosch, South Africa
4   Biostatistics Unit, Institute of Crop Science, University of Hohenheim, 70593 Stuttgart, Germany
5   Department of Meteorology, University of Nairobi, P.O Box 30197, Nairobi, Kenya
*   Correspondence: lydiaolaka@uonbi.ac.ke

**Abstract:** Rivers in the Lake Victoria Basin support a multitude of ecosystem services, and the economies of the riparian countries (Kenya, Tanzania, Uganda, Rwanda, and Burundi) rely on their discharge, but projections of their future discharges under various climate change scenarios are not available. Here, we apply Vector Autoregressive Moving Average models with eXogenous variables (VARMAX) statistical models to project hydrological discharge for 23 river catchments for the 2015–2100 period, under three representative concentration pathways (RCPs), namely RCPs 2.6, 4.5, and 8.5. We show an intensification of future annual rainfall by 25% in the eastern and 5–10% in the western part of the basin. At higher emission scenarios, the October to December season receives more rainfall than the March to May season. Temperature projections show a substantial increase in the mean annual minimum temperature by 1.3–4.5 °C and warming in the colder season (June to September) by 1.7–2.9 °C under RCP 4.5 and 4.9 °C under RCP 8.5 by 2085. Variability in future river discharge ranges from 5–267%, increases with emission intensity, and is the highest in rivers in the southern and south eastern parts of the basin. The flow trajectories reveal no systematic trends but suggest marked inter-annual variation, primarily in the timing and magnitude of discharge peaks and lows. The projections imply the need for coordinated transboundary river management in the future.

**Keywords:** hydrological flows; VARMAX model; water sector; Lake Victoria Basin; East Africa; RCPs 2.6, 4.5 and 8.5

---

## 1. Introduction

Most climate models project increasing rainfall over East Africa (EA) in the coming decades, however, the long rainy season, March to May (MAM), has been experiencing devastating droughts whereas the October to December (OND) season's rainfall is increasing [1–3]. Future hydrological flows and freshwater supply, which are important to the economies of the East African countries, have not yet been comprehensively analyzed due to the scarcity and incompleteness of relevant data. The dependence of people on rivers for a variety of ecosystem services make them particularly vulnerable to impacts of climate change [4–6]. In recent decades, the per capita water availability in many countries globally has declined largely due to increasing demand, related to growth in the human population, industrialization, and urbanization. This trend is likely to continue in the coming decades [7–10] and it

is anticipated that climate change will continue to affect river hydrology and ecology through changes in rainfall distribution, soil moisture, river flows, and groundwater levels [11,12]. Recent fluctuations in river levels have had adverse impacts on the social, economic, and environmental well-being of many African communities [13,14]. Major changes in mean river discharge can have devastating impacts, particularly in Africa.

Rainfall deficits of between 7% and 29% in East Africa between 1961 to 2010 led to sharp reductions in agricultural output and employment and also resulted in significant losses in Gross Domestic Product (GDP) [15]. Similar economic losses associated with drought conditions have occurred in West Africa, Australia, California, and Southern Africa [16–20]. Such losses emphasize the economic connection of climate and hydrology to water and sanitation, agriculture, fisheries, and energy sectors. Consequently, multiscalar present and future climate change studies are valuable for advancing scientific understanding and providing information for decision making in adaptation and mitigation strategies to deal with widening variability in river flows [21,22].

Other threats to freshwater that are exacerbated by climate change include increased river siltation resulting from high soil erosion in the basin, recurrent destructive floods in the low-lying areas, riparian land encroachment, degradation of river banks, eutrophication, and proliferation of the invasive water hyacinth [23,24]. Increasing intensity and frequency of extreme climatic events pose additional threats to the future ecological and community well-being in the Lake Victoria Basin (LVB) [25].

Long term climate records have helped establish the relationship between Lake Victoria levels and the regional climate [26–28], which is influenced, in turn, by regional and hemispheric phenomena, such as the El Niño–Southern Oscillation (ENSO) and the Indian Ocean Dipole (IOD) [29–32]. However, due to pronounced topographical gradients and influence of the large water body in Lake Victoria, the climate of the Lake Victoria Basin (LVB) varies widely over space and time [30,33,34]. Thus, climate variability and change are expected to impact on water resources and rivers in the future decades. Some studies have projected a 10–20% increase in stream flows in East Africa by 2040 [35,36] but did not consider spatial variation in river catchment flows. Relatively little is known on the impact of projected climate change on the hydrology of the specific rivers draining into Lake Victoria under different future anthropogenic emission scenarios. Moreover, transboundary rivers in the basin create hydrological, social, and economic interdependencies between societies, requiring transboundary water management [37]. Thus, sub-basin level analyses are needed because responses to climate change may vary in space across the basin. These insights are fundamental for the development of scientifically sound regional adaptation and mitigation strategies and also to achieve sustainable development goals targets related to climate change impacts, clean water and sanitation, conservation, and protection of life on land and water within the LVB.

Hydrological models used for streamflow projections are usually data intensive and require parameters of the physical characteristics of catchments and hydroclimatic variables, which are lacking for most EA countries. To overcome this limitation, we use the vector autoregressive moving average processes (VARMAX) model to represent the dynamic relationships between the historic river discharge series and rainfall and temperature series and project the future river flow discharge. This approach only requires long-term historical climate and discharge data. Accordingly, our main aim is to establish the future spatiotemporal changes in the hydrology of rivers draining into the LVB for the period of 2015–2100 in response to three emission scenarios using the VARMAX model. The scenarios are distinguished by three emission levels (scenarios) of increasing intensity. This would form the basis for the development of adaptation and mitigation measures in East Africa. We use historic climate records spanning 1971 to 2000 to represent the historical baseline or reference for assessing future changes.

## 2. Materials and Methods

*2.1. Study Area*

Lake Victoria is the second largest freshwater lake in the world with a lake surface area of 68,000 km$^2$ and a total basin area of 250,000 km$^2$. This lake straddles the equator and its shoreline is shared directly by three East African countries (Kenya, Uganda, and Tanzania), making it an important transboundary lake. The lake is recharged by direct rainfall and 23 rivers flowing from five east African countries (Burundi, Rwanda, Kenya, Tanzania, and Uganda). There are only two trans-boundary rivers into the lake (i.e., Kagera, which flows through four countries: Rwanda, Burundi, Uganda, and Tanzania; and the Mara River, which flows through Kenya and Tanzania). Of the remaining rivers, 8 are in Kenya (North Awach, South Awach, Gucha Migori, Nyando, Nzoia, Sio, Sondu, and Yala), 10 in Tanzania (Biharamulo, East shore streams, Gurumeti, Issanga, Magogo Moame, Mbalageti, Nyashishi, and West shore streams), and only 3 rivers in Uganda (Bukora river, North shore streams, and Katonga rivers) (Figure 1) [38]. Outflow from the lake is via the white Nile River to the Mediterranean; the Nile river supports the livelihoods of about 300 million people, as it is one of the two major sources of the Nile [39,40]. The mean annual rainfall ranges from 1350 mm in Kenya to a maximum of 2400 mm in the Ugandan part of the basin. Precipitation is the largest contributor of water (80%) to the lake [41].

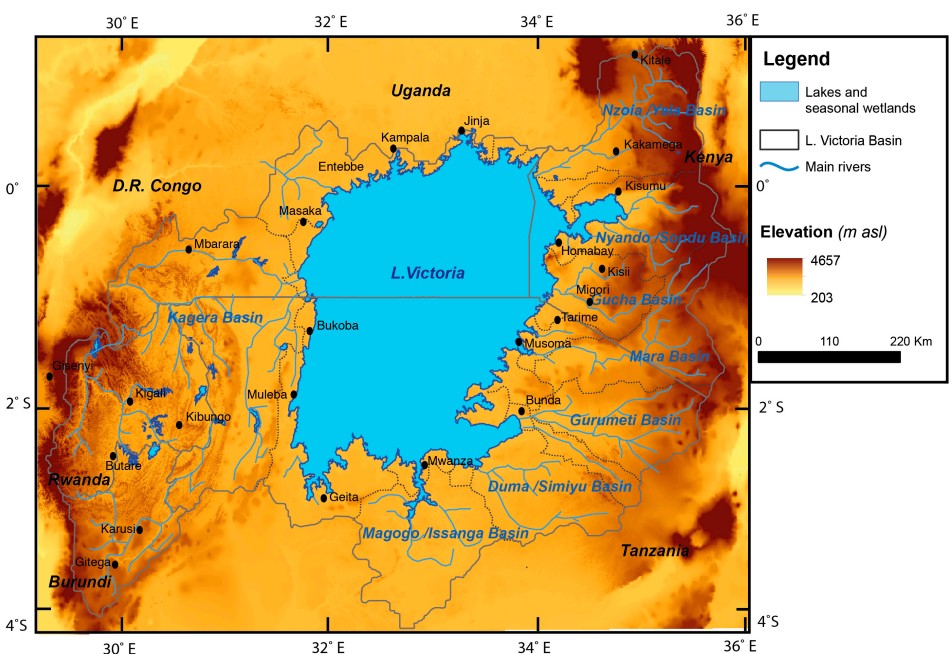

**Figure 1.** Rivers of the Lake Victoria Basin used in the projection studies.

These rivers are important for domestic, industrial, and irrigation water supply and also for supporting navigation, tourism, and energy production. About 800 MW of power is produced from these five hydroelectric power stations in the LVB: At the outlet from Lake Victoria into the Nile in Jinja (Nalubaale, Kiira, and Bujagali hydroelectric power stations), Sondu Miriu on Sondu Miriu River in Kenya, and the Rusumo Falls power station, which is under construction on the Kagera River. The livelihoods of about 35 million people, with an estimated growth rate of between 2% and 3.4% per annum (UNEP, 2006), depend on the LVB. The population density in the region is among the highest in the world, averaging more than 500 persons/km$^2$ and exceeding 1200 persons/km$^2$ in some parts. Due to intense use of natural resources, the rivers' headwaters in Kenya (the Mau Escarpment) and Rwanda-Burundi (Bugesera region and Kagera) are experiencing mounting pressures from deforestation and siltation [42].

Fresh water availability across EAC countries is heterogeneous due to both climatic and non-climatic factors, such as the freshwater supply infrastructure and contamination. Accordingly, Burundi is classified as water scarce, Rwanda and Kenya are water stressed, while Uganda and Tanzania have relatively high water sufficiency. This unequal distribution of water is a limiting factor for both agricultural and industrial growth, a potential source of conflict between the five countries, and a source of competition between rural community needs and the urban industrial sector. Climate change and increasing variability is likely to exacerbate these factors.

## 2.2. Historical Rainfall, Minimum and Maximum Temperatures, and River Flow Data

The historical rainfall data are based on the gridded observation/satellite blended Climate Hazards group Infrared Precipitation with Station data (CHIRPS). CHIRPS is a 30+ year quasi-global rainfall dataset spanning 50° S–50° N (including all longitudes) starting in 1981. CHIRPS incorporates 0.05° resolution satellite imagery with in-situ station data to create gridded rainfall time series for trend analysis and seasonal drought monitoring. The temperature data were sourced from the USGS FEWSnet data portal (https://earlywarning.usgs.gov/fews/software-tools/20 [43]). The GeoCLIM software tool was used to extract the historical rainfall, and minimum and maximum temperature data for the period of 1951–2005 for the entire LVB. Details on GeoCLIM can be found at http://chg-wiki.geog.ucsb.edu/wiki/GeoCLIM [44].

The annual river discharge data for all the 23 rivers draining into LVB during 1950 to 2000 were acquired from the Nile Basin Initiative (NBI) database. The discharge for the 23 rivers during 1950 to 2000 and the total annual rainfall for LVB for the Kenyan, Tanzanian, and Ugandan sections for the same period are provided in Table S1.

## 2.3. Climate Projections

The IPCC (2013) recommended a series of scenarios known as Representative Concentration Pathways (RCPs) representing the full bandwidth of possible future emission trajectories. Here, we use three RCPs: RCP 2.6, RCP 4.5, and RCP 8.5 to project climate scenarios up to 2085. The RCP 2.6 scenario represents an optimistic projection characterized by very low concentration and emission levels of greenhouse gases, with a medium rate of population growth, and the radiative forcing peaks at 3 Wm$^{-2}$ in the 2050s before a decrease in 2100. The RCP 4.5 scenario stabilizes radiative forcing at 4.5 Wm$^{-2}$ in the year 2100 without ever exceeding that value. This scenario assumes that climate policies, particularly the introduction of a set of global greenhouse gas emission policies, are invoked to achieve the goal of limiting emissions, concentrations, and radiative forcing. The RCP 8.5 scenario represents a pessimistic projection with high levels of concentrations and emissions of greenhouse gases, a high rate of population growth, and radiative forcing reaching 8.5 Wm$^{-2}$ by 2100. This scenario assumes that no climate change policies are implemented [45–47].

We used the simulated data from the Rossby Center Regional Atmospheric Model (RCA4) driven by the Earth system version of the Max Planck Institute for Meteorology (MPI-ESM-LR) coupled global climate model from the on-going Coordinated Regional Climate Downscaling Experiment (CORDEX) project. The goal of the Regional Climate Model (RCM) program is to advance the predictive understanding of the Earth's climate by focusing on scientific analysis of the dominant sets of governing processes that describe climate change on regional scales. The model was integrated into the CORDEX-Africa domain, with a horizontal grid spacing of 0.44 degrees—translating to a 50 × 50 km grid [48,49]. The historical simulations were forced by observed natural and anthropogenic atmospheric composition covering the period from 1951 until 2005, whereas the projections (2006–2085) were forced by the three RCPs.

The projected changes in rainfall and temperature for the three scenarios were based on three time slices, 2030s (2016–2045), 2050s (2036–2065), and 2070s (2055–2085), to provide information on the expected magnitude of the climate response over each time window [50]. The period of 1971–2000 is considered as a reference for the present climate. The projected climate change signals for each time

window are calculated as the difference between the future time window and the reference period. Projections are made for both the annual and seasonal components of rainfall, and the minimum and maximum temperatures. Annual rainfall is the total rainfall over January to December of each year. Annual minimum (maximum) temperature is the average of the daily minimum (maximum) temperature over January to December. Rainfall in the LVB was partitioned into distinct seasons, corresponding to the main long rains (MAM), short rains (OND), and the dry season rains (June to September—JJAS) [51–53]. The total annual rainfall for 2015–2085 projected under the RCP 2.6, 4.5, and 8.5 emission scenarios for sections of the basin falling in each of the five LVB states are provided in Table S2.

## 2.4. Statistical Analysis and Projection of River Discharge

Statistical methods for time series analysis and projections were used to analyze temporal variation in the historic observations of river discharge. The historic observations of river discharge (response variables) were first related to rainfall, and the minimum and maximum temperatures (predictor variables). As explanatory variables involving rainfall, we applied a lag of 0 to 5 years in the total annual rainfall and calculated six moving averages of the annual rainfall spanning time windows of lengths 1 to 5 years. The natural logarithm of discharge, LN (river discharge +1), was then related separately to each of the six lags and six moving averages of annual rainfall using generalized linear models assuming a Gaussian error distribution and the canonical identity link function. Each rainfall component was divided by the long-term average annual rainfall to obtain deviations from the long-term mean. The best supported rainfall component and the functional form of the relationship of this component with river discharge was selected using the corrected Akaike information criterion (AICC, Table S4). The linear model had the strongest support even though we also considered several contending nonlinear models (e.g., quadratic and semi-parametric regression models). The relationships we established for the historic river discharge series were then used to project the likely future trajectories of the discharge series using time series forecasting models.

The primary goal was to use univariate and multivariate time series analysis to model variation in the river discharge series and use the model to forecast future values of the series. A secondary goal was to obtain smoothed trend patterns and associated uncertainty measures, impute missing values through interpolation, and model the structure of the series. Accordingly, we carefully considered and modeled several characteristics of the river discharge time series, including cycles, trend, serial autocorrelation, and moving averages.

We used the vector autoregressive moving average processes (VARMAX) model to characterize the dynamic relationships between the river discharge series and the selected rainfall and minimum and maximum temperature components, and to forecast the discharge series under each of the three scenarios. The VARMAX model we used allowed for: (1) Modelling of multiple time series simultaneously, (2) accounting for relationships among the individual river discharge component series with current and past values of the other series, (3) feedback and cross-correlated predictor series, (4) cointegration of component time series to achieve stationarity, (5) seasonality, (6) autoregressive errors, (7) moving average errors, (8) mixed autorergressive and moving average errors, (9) lagged values of the explanatory series, and (10) unequal or heteroscedastic covariances for the residuals and more.

The VARMAX model can be represented in various forms, including in state space, dynamic simultaneous equation, or dynamic structural equations forms and allows representation of distributed lags in the explanatory (predictor) variables. For example, we can simultaneously relate river discharge in year $t$ to river discharge in year $t - 1$, $t - 2$, annual rainfall in year $t$, $t - 1$, $t - 2$, and minimum and maximum temperatures in years $t$, $t - 1$, and $t - 2$. Univariate autoregressive moving average models were used to relate river discharge to rainfall and minimum and maximum temperatures as the predictor variables. The univariate autoregressive moving average models were fitted separately to each of the 23 rivers. Models with multiple lags in rainfall or distributed lag models were also

considered. Various lags in rainfall and minimum and maximum temperatures were tested so that the models could be characterized as autoregressive and moving-average multiple regression with distributed lags. The significance of seasonal (wet and dry season) deterministic terms for rainfall and temperature time series data were similarly tested. The dead-start models that exclude present (current) values of the explanatory variables were examined and tested for heteroscedasticity in residuals by allowing for GARCH-type (generalized autoregressive conditional heteroscedasticity) conditional heteroscedasticity of residuals. Several Information-theoretic model selection criteria were used to determine the AR (autoregressive) and MA (moving average) orders of the models. The specific criteria we used were the Akaike information criterion (AIC), the corrected AIC (AICC), Hannan–Quinn (HQ) criterion, Schwarz Bayesian criterion (SBC), also known as Bayesian information criterion (BIC), and the final prediction error (FPE). As additional AR order identification aids, we used partial cross-correlations for the response variable, Yule–Walker estimates, partial autoregressive coefficients, and partial canonical correlations. Parameters of the selected full models were estimated using the maximum likelihood (ML) method. Roots of the characteristic functions for both the AR and MA parts (eigenvalues) were evaluated for proximity to the unit circle to infer evidence for stationarity of the AR process and inevitability of the MA process in the response series (Table S5).

The adequacy of the selected models was assessed using various diagnostic tools. The specific diagnostic tools we used are the following: (1) Durbin–Watson (DW) test for first-order autocorrelation in the residuals (Table S4); (2) Jarque–Bera normality test for determining whether the model residuals represent a white noise process by testing the null hypothesis that the residuals are normally distributed (Table S6); (3) F- tests for autoregressive conditional heteroscedastic (ARCH) disturbances in the residuals, this F statistic tests the null hypothesis that the residuals have equal covariances (Table S6); (4) F tests for AR disturbance computed from the residuals of the univariate AR(1), AR(1,2), AR(1,2,3), and AR(1,2,3,4) models to test the null hypothesis that the residuals are uncorrelated (Table S7); and (5) Portmanteau test for cross correlations of residuals at various lags. Final forecasts and their 95% confidence intervals were then produced for the river discharge series for lead times running up to 2100 (Table S8). Univariate model ANOVA diagnostics show that the selected model for each river is significant and has an $r^2$ ranging between 0.35 and 0.79 (Table S9).

All the established regression relationships were also visualized with the aid of various graphical plots. Model selection showed that a VARMAX (2,2,4) model was appropriate and adequate for all the 23 series, where VARMAX ($p,q,s$) means that $p = 2$ is the order of autoregression, $q = 2$ is the order of the moving average errors, and $s = 4$ is the number of lags in annual rainfall. The VARMAX ($p,q,s$) model we used to forecast the future dynamics of river discharge can thus be expressed as:

$$R_t = \sum_{j=1}^{p=2} \Phi_j R_{t-j} + \sum_{j=0}^{s=4} \Omega_j^* x_{t-j} + \epsilon_t - \sum_{j=1}^{q=2} \Omega_j \epsilon_{t-j}, \tag{1}$$

where $R_t$ is the natural logarithm of the average river discharge in year $t$, $x_t = (\text{Rain}_{t-0}, \dots, \text{Rain}_{t-4})^T$ are the total annual rainfall lagged by 0 to 4 years, with each rainfall lag divided by the long-term mean annual rainfall. $\epsilon_t$ is the vector of the white noise process (errors) in $R_t$. It is assumed that $E(\epsilon_t) = 0$, $E(\epsilon_t \epsilon_t^T) = \Sigma$ and $E(\epsilon_t \epsilon_u^T)$ for $t \neq u$. We further assumed that $p$ and $q$ are each equal to 2 whereas $s$ was set to equal 4. The regression relationships in model (1) were used to project the likely future trajectories of the river discharge rate using the projected rainfall under each of the three climate scenarios for each river and the model parameters estimates in Table S10.

The historical (1950–2014) and projected (2015–2100) discharge, the associated residuals, standard errors, and pointwise lower and upper 95% confidence limits for each of the 23 rivers in the LVB under the RCP 2.6, 4.5, and 8.5 emission scenarios are provided in Table S2.

## 3. Results

### *3.1. Relationships between Historical Discharge and Rainfall*

The average annual discharge for each of the 23 rivers for the historic period (1950–2000) is positively linearly correlated with the total annual rainfall for the same period (Figure S1a–c) even though the strength of the correlation varies across rivers from 0.21 to 0.83 (Table S11). As a result, the linear model had stronger support than all the nonlinear models (Figure S1a–c). Seventeen rivers having a significantly high and positive correlation (0.50–0.83) with rainfall occur in the eastern and southern part of the basin (Gucha-Migori, Nyando, Sio, S. Awacha, North Awach, Nzoia, Sondu, Yala, Issanga, Mara, S. shore streams, E. Shore streams, Kagera, Mbalageti, Simiyu, Magogo Moame, and Nyashishi). In contrast, rivers in the west have a low to moderate but significant positive correlation (0.31–0.49) with rainfall (Biharamulo, Grumeti, W. shore streams, Bukora, and N. Shore streams). Katonga in the west has the least and the only insignificant positive correlation (0.21) with rainfall, reflecting a notably high variation in discharge (Figure S1a–c, Table S11). For all the rivers except Katonga, the slope of the linear regression of discharge on the total annual rainfall is positive and significant (Table 1). The minimum and maximum temperatures were also used as predictors in the exact same way as rainfall but had little support and thus were not used to project river discharge.

### *3.2. Climate Projections for the Lake Victoria Basin for 2030–2085*

#### 3.2.1. Rainfall

The annual and seasonal rainfall components are projected to change differentially in space over the LVB and across the three scenarios between 2015 and 2085. During the 2030s, the annual rainfall will increase by 5–10% over most of the basin under RCP 4.5 but mainly over the eastern section of the basin under RCPs 2.6 and 8.5 (Figure 2). In contrast, the long rains (MAM) will reduce by 5–10% over most of the lake and on the western and eastern sections of the basin under RCP 2.6. The MAM season rains also show a similar pattern of decrease, to a lesser extent, under RCPs 4.5 and 8.5, and a 5–10% increase over most of the south eastern and north western sections of the basin, especially under RCP 4.5 (Figure 2). In contrast, the short rains (OND) will increase over most of the lake and basin and by up to 25% on the eastern headwaters of the basin under all three scenarios. The long dry season rains (JJAS) will reduce by 5–50% over most of the basin under all the three scenarios but increase by 5–10% over the lake, northern, and north eastern sections of the basin, most noticeably under RCP 4.5 (Figure 2).

The projected changes in the annual and seasonal rainfall for the 2050s are largely similar to those for the 2030s with minor exceptions. Thus, the annual rainfall is projected to increase by 5–25% over the eastern, southern, and north central sections of the basin under all three scenarios. However, the central portion of the lake is projected to have less annual rainfall in the 2050s than either the base period or the 2030s under all three scenarios (Figure 3). The projections also show contrasting trajectories for the different seasonal rainfall components in the 2050s. The MAM season rains will reduce by up to 25% in the central, western, and north eastern portions of the basin under all three scenarios. This contrasts with the OND season rains projected to increase over most of the basin and by up to 25% on its eastern headwaters, spanning a larger spatial extent than for the 2030s. The JJAS season rains also show a similar general pattern to that for the 2030s but will be drier over most of the basin (Figure 3).

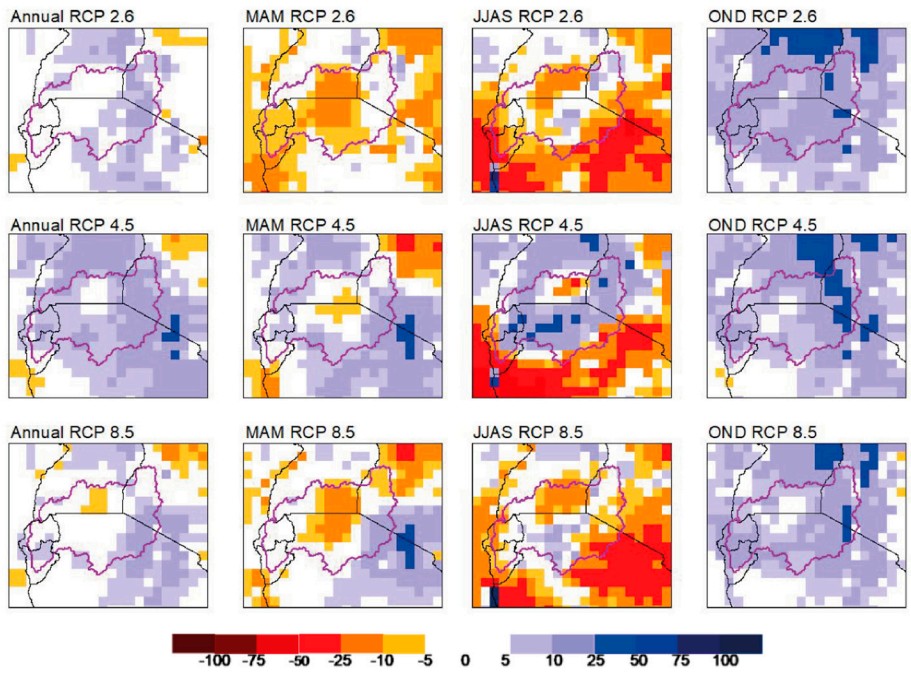

Fig 2

**Figure 2.** Projected changes in the annual, March–April–May (MAM), June–July–August–September (JJAS), and October–November–December (OND) rainfall components over the Lake Victoria Basin by the 2030s. Each row corresponds to the emission scenarios of RCP 2.6, RCP 4.5, and RCP 8.5.

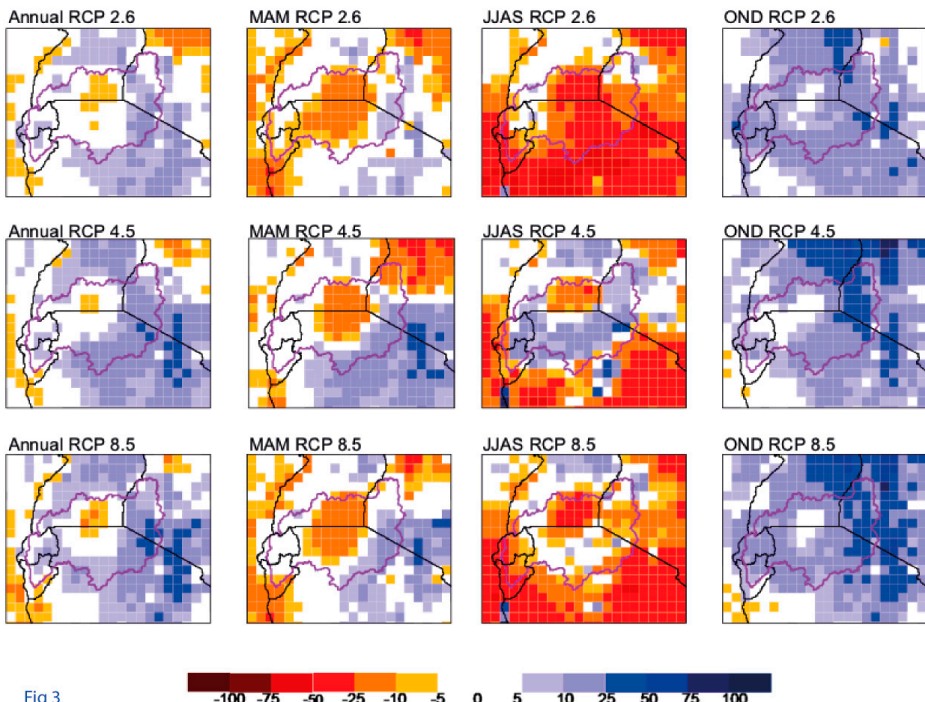

Fig 3

**Figure 3.** Projected changes in annual, March–April–May (MAM), June–July–August–September (JJAS), and October–November–December (OND) components over the Lake Victoria Basin by the 2050s. Each row corresponds to the emission scenarios of RCP 2.6, RCP 4.5, and RCP 8.5.

The changes projected for the annual and seasonal rainfall for the 2070s broadly mirror those for the 2030s and 2050s and differ mainly in their spatial extent and magnitude. Thus, the annual

rainfall projected for the 2070s is lower than those for the 2030s and 2050s only under the low emission scenario (RCP 2.6). At the higher emission scenarios (RCPs 4.5 and 8.5), the annual rainfall will have two opposing patterns: An increase of up to 25% over a large eastern portion of the basin and a 5–10% decline in rainfall over the lake area (Figure 4). The MAM season rains are projected to reduce by up to 25% over the central portion but to increase by 5% over the south eastern portions of the basin under RCP 8.5. By contrast, the OND season rains will increase by up to 25%, mostly in the eastern, south eastern, and north central sections of the basin under RCP 8.5 (Figure 4). The JJAS season rains will be 5–50% lower than currently over most of the basin, particularly in its southern section. The JJAS season rains will decrease over most of the basin under RCPs 2.6 and 8.5 but increase by 5–10% over most of the lake and in the north central section of the basin under RCP 4.5.

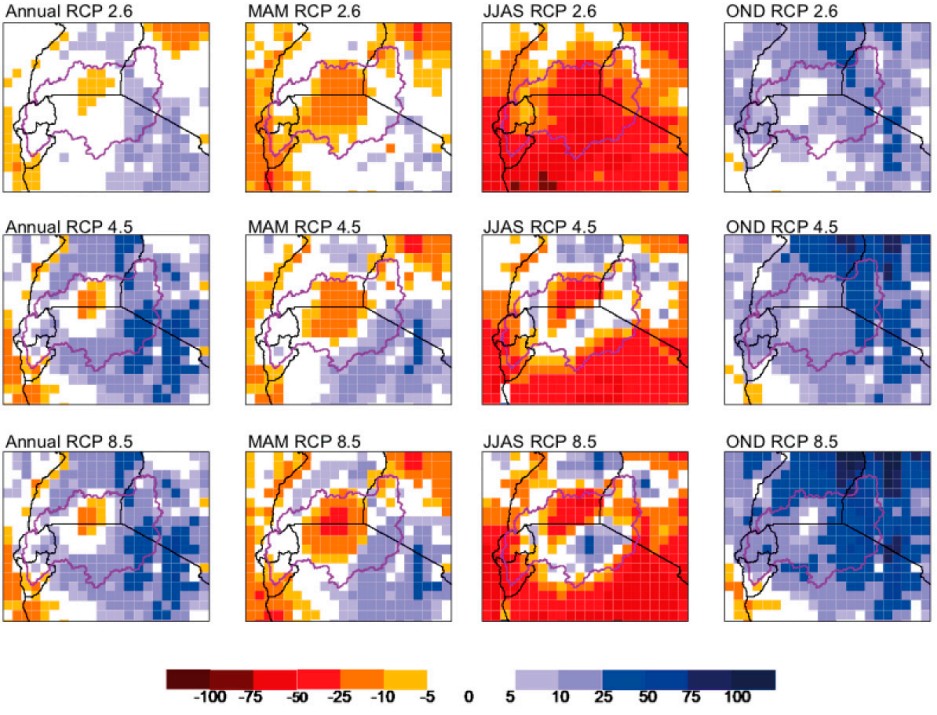

Fig 4

**Figure 4.** Projected changes in the annual, March–April–May (MAM), June–July–August–September (JJAS), and October–November–December (OND) rainfall components over the Lake Victoria Basin by the 2070s. Each row corresponds to the emission scenarios of RCP 2.6, RCP 4.5, and RCP 8.5.

### 3.2.2. Maximum and Minimum Temperatures

The projected changes in the mean annual, maximum (Tmax) and minimum (Tmin) temperatures vary across both the basin and the three scenarios in the 2030s, 2050s, and 2070s (Figures S2–S4). By the 2030s, the mean annual maximum temperature will increase throughout the LVB by about 1.0–1.5 °C under RCPs 2.6 to 8.5. For the OND season, Tmax will increase over the central part of the basin by 0.5–0.7 °C under RCPs 2.6 and 4.5 and over the western part by 0.9 °C under RCP 8.5. Slightly higher increases in Tmax are projected for the MAM than for the OND season by the 2030s. Specifically, in the MAM season, Tmax is projected to be 0.9–1.7 °C warmer by the 2030s under all three scenarios. Moreover, the increase in Tmax will be homogeneous over the basin under RCP 2.6 but will be higher on the western than the eastern section of the basin under RCPs 4.5 and 8.5. For the JJAS season, Tmax is projected to increase by 1.2–1.6 °C under all three scenarios. This increase will be greater in the southwestern part than in the rest of the basin by the 2030s under RCP 2.6. However, Tmax is projected to increase by 1.3 °C in the whole basin under RCP 4.5 and to increase more in the higher elevation areas (1.6 °C) than in the central part (1 °C) of the basin under RCP 8.5.

The mean annual minimum temperature is projected to increase over the basin by 1.2–1.9 °C by the 2030s under all three scenarios (Figure S2). The spatial extent of the basin over which Tmin will increase (by >1.5 °C) will widen with emission intensity from RCP 2.6 to 8.5. Across seasons, Tmin is projected to increase by 1.4–2 °C under all three scenarios and by a smaller margin in the JJAS season than the MAM and OND seasons. Tmin is also projected to be warmer (>1.7 °C) over a larger area of the basin under RCPs 4.5 and 8.5 than 2.6. The projected warming in Tmin will be greater on the margins than at the center of the basin. Under RCP 2.6, Tmin is projected to increase by 1.0–1.5 °C, mostly over the central and southern parts of the basin. In the MAM and OND seasons, Tmin will increase, but spatially unevenly, by 1.0–1.7 °C under all three scenarios. The area of the basin over which Tmin will increase stretches from east to west and enlarges with emission intensity from RCP 2.6 to 8.5.

By the 2050s, the mean annual maximum temperature is projected to increase under RCPs 2.6 (1.2–1.4 °C), 4.5 (1.6–1.8 °C), and 8.5 (2.1–2.3 °C). The increase in Tmax is homogenous across the LVB under RCP 2.6, but it is higher over the rest of the basin than at its center under RCPs 4.5 and 8.5. Tmax is also anticipated to be warmer (1.4–2.7 °C) in the MAM and JJAS seasons than in the OND season (0.8–1.8 °C) under all three scenarios. Tmax will increase from east to west and with increasing emission intensity (from RCP 2.6 to 8.5 (Figure S3)). By the 2050s, the increase in Tmax is smaller than that in Tmin under RCPs 2.6 (1.5–1.7 °C), 4.5 (2.0–2.4 °C), and 8.5 (2.5–3 °C). The warming in Tmin is greater on the western than the eastern part of the basin for all three scenarios. Tmin is also projected to become warmer in the MAM (1.3–2.7 °C), JJAS (1.7–3.0 °C), and OND (1.2–3.0 °C) seasons for all three scenarios. Tmin will also become warmer in the JJAS season over an extensive area of the basin under all three RCP scenarios.

By the 2070s, the mean annual maximum temperature is projected to warm up under RCPs 2.6 (1.1–1.4 °C), 4.5 (1.8–2.1 °C), and 8.5 (3.2–3.6 °C) (Figure S4a). As a result, Tmax will be warmer in the 2070s than the 2050s under RCPs 4.5 and 8.5. The anticipated increase in Tmax by the 2070s is spatially homogenous over the basin under RCP 2.6, greater over the western than the eastern part of the basin under RCP 4.5, and smaller over the center than elsewhere in the basin under RCP 8.5. The projected increase in mean Tmax also varies across seasons and for the MAM season across RCPs 2.6 (1.3–1.7 °C), 4.5 (2.0–2.4 °C), and 8.5 (2.3–2.5 °C). For the JJAS season, the projected rise in Tmax also varies across RCPs 2.6 (1.5–1.7 °C), 4.5 (1.9–2.1 °C), and 8.5 (3.9–4.3 °C). For the OND season, a lower increase in Tmax is expected across RCPs 2.6 (0.8 °C), 4.5 (1.3–1.6 °C), and 8.5 (2.2–2.7 °C) (Figure S4). The expected warming in Tmax varies spatially and is higher for the western than the eastern section of the basin under RCPs 2.6 and 4.5. However, under RCP 8.5, Tmax will be warmer over an extensive area around the basin than over the lake area. The increase in Tmin projected for the 2070s also varies across RCPs 2.6 (1.5 °C), 4.5 (2.3–2.8 °C), and 8.5 (3.8–4.5 °C). Tmin is also projected to become warmer in the western than the eastern part of the basin. Seasonally, Tmin will be 1.6 to 5 °C warmer in the JJAS than the MAM or the OND season under all scenarios (Figure S4a). Finally, Tmin will be warmer over the lake area and in the western than the eastern part of the basin under all three scenarios.

### 3.3. Stream Flow Projections for LVB Rivers for 2015–2100

To quantify the expected impact of climate change on hydrological or annual river discharge, we used the deviation of the projected river flow rate for each scenario and period (2015–2030, 2031–2050, 2051–2070, and 2071–2100) combination from the corresponding historic baseline rate. The summary and descriptive statistics for the projected river discharge rate are used to compare changes in discharge across scenarios and periods in Table 1.

### 3.3.1. Rivers on the Eastern Part of the L. Victoria Basin

The river discharge projections for the eight rivers draining the eastern part of the basin (Kenya side—Gucha Migori, North Awach, Nyando, Nzoia, Sio, Sondu, South Awach, and Yala) are summarized in Figure 5. River discharge projections for the coming decades for the eight rivers

draining the eastern part of the basin have no apparent systematic trend except for one (River. Nyando) river. At lower and intermediate emission scenarios (RCP 2.6 and 4.5 scenarios), the average discharge level for Nyando river will be higher than for the base period throughout the 21st century, indicating upward level shifts. A slight upward trend is also apparent for the RCP 2.6 and 4.5 scenarios from 2071 to 2100. However, discharge for Nyando river at higher emission scenarios (RCP 8.5) will be lower than the average discharge for the base period initially (2015–2070) but will increase between 2071 and 2100 (Figure 5). The coefficient of variation for the rivers in the eastern part (Table 1) will range between 20% and 260% during 2015 and 2100 for the three emission scenarios. The highest %CV (46–260%) will be for river Nyando, ranging from 46–154% in the 2030s and reaching a peak (119–267%) in the 2050s before reducing to 110–136% in the 2070s and to 114–160% in the 2100s. The least %CV during the 2015–2100 period will be for S. Awach river (20–50%). In most cases, the %CV for the 21st century will increase with increasing emission levels (RCP 8.5 > RCP 4.5 > RCP 2.6).

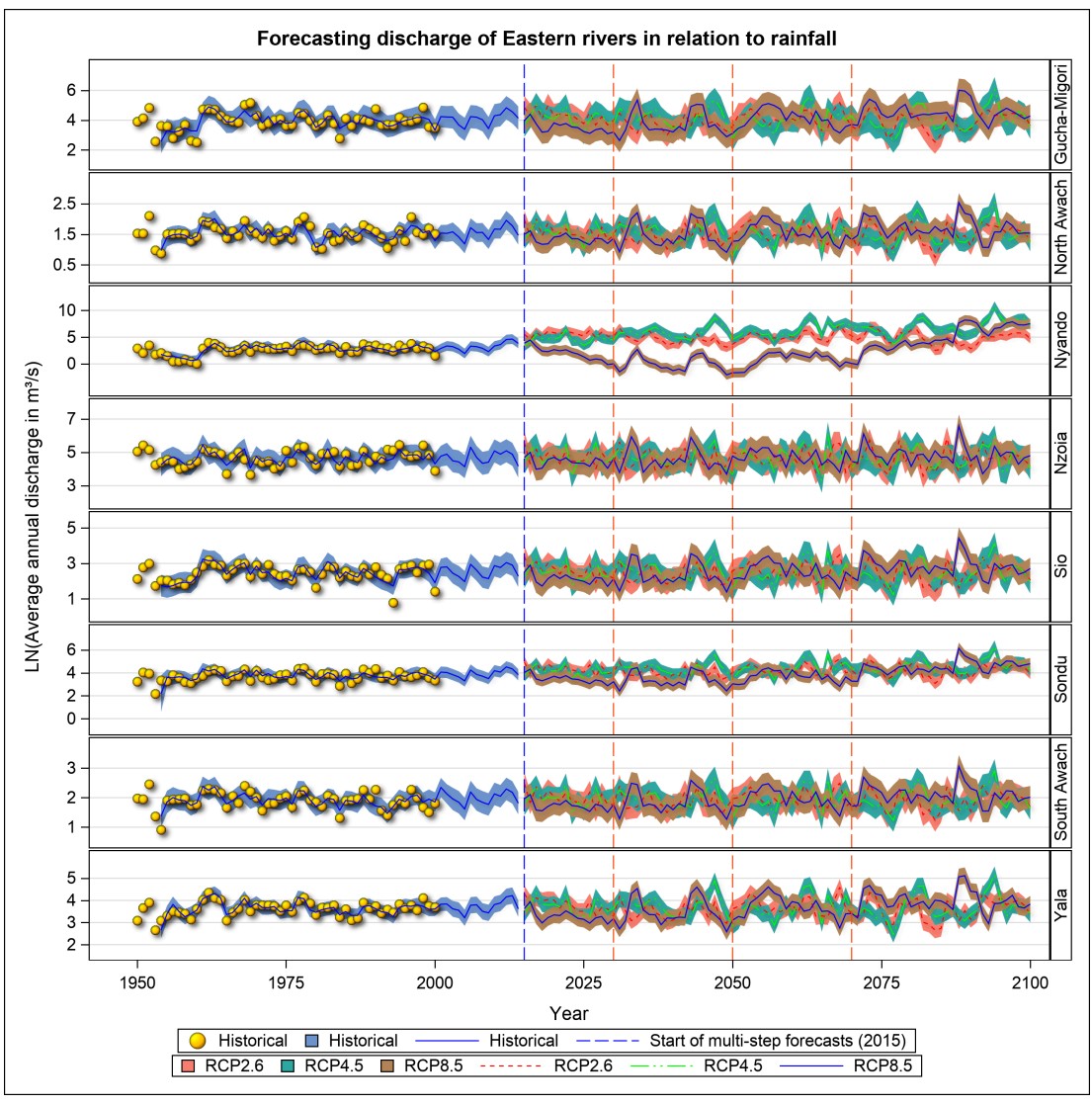

(**a**)

**Figure 5.** *Cont.*

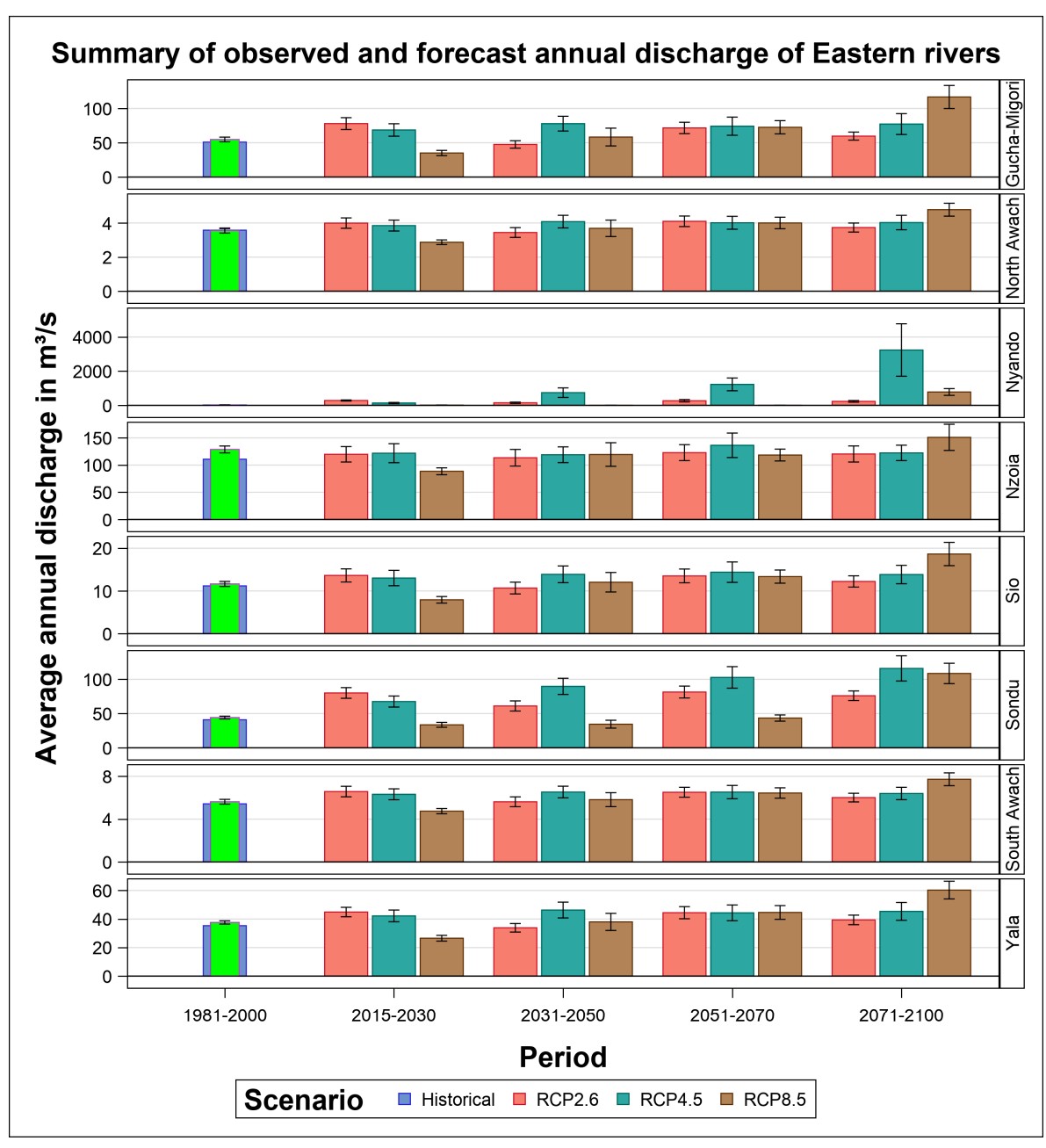

(**b**)

**Figure 5.** (**a**) Projected average annual river discharge in relation to rainfall and (**b**) summary of the observed (historical) and projected average annual discharge for rivers flowing into the eastern part of the Lake Victoria Basin for the 2030s, 2050s, and 2070s under the RCP 2.6, 4.5, and 8.5 scenarios. The bright green bar is the average of the observed historic series while the wider medium slate blue bar is the value for the historic series predicted by the fitted model.

Despite the absence of systematic trends, the discharge trajectories projected suggest marked inter-annual variation regardless of the scenario. This variation is characterized by pronounced quasi-periodic oscillations. The oscillations have time-varying amplitudes and phases and differ between scenarios primarily in the timing and magnitude of discharge peaks and troughs. Notably, the peaks and troughs are most pronounced for RCP 8.5, intermediate for RCP 4.5, and least for RCP 2.6 compared to historical flows. There are more extended periods of low discharge under RCP 8.5

than under the other two scenarios, suggesting more protracted periods of water scarcity, on average, under this scenario (Figure 5a,b).

In the 2015–2030 period, the mean monthly discharge projected for each of these rivers will be consistently highest under RCP 2.6, intermediate under RCP 4.5, and lowest under RCP 8.5. The highest discharge is expected for Nyando and the lowest for North Awach river (Figure 5a,b). The expected discharge trajectories under RCPs 2.6 and 4.5 are comparable but substantially higher than that under RCP 8.5. The average discharge is most variable under RCP 4.5, middling under RCP 2.6, and least variable under RCP 8.5 (Table 1).

For the 2031–2050 period, the projected average discharge is highest for RCP 4.5, intermediate for RCP 8.5, and lowest for RCP 2.6. The only exceptions are Sondu and Nyando rivers, for which the average discharge is higher under RCP 2.6 than RCP 8.5 (Figure 5a). The projected average flow rate is most variable under RCP 8.5, intermediate under RCP 4.5, and least variable under RCP 2.6 (Table 1).

By the 2051–2070 period, the projected average river flows are comparable across all but two rivers (Nyando and Sondu) under all three scenarios. Both rivers flow from the eastern part of LVB and they attain the highest, medium, and lowest average flow rates under RCPs 4.5, 2.6, and 8.5, respectively (Figure 5a,b). The variances in the river flow rates are mainly similar under RCPs 2.6 and 4.5 but smaller than that under RCP 8.5, (Table 1). The average discharge is also consistently highest for RCP 8.5, medium for RCP 4.5, and least for RCP 2.6 for the 2071–2100 period. Sondu river deviates from this pattern and has the highest, medium, and least flow rates under RCPs 4.5, 8.5, and 2.6, respectively (Figure 5a,b). Nyando river has the most variable discharge across all the periods and scenarios (Table 1). Further, the average discharge for the base period (1981–2000) is generally smaller than, or comparable to, the projected average discharge for all the periods under RCPs 2.6 and 4.5. For the Sondu river the projected discharge under RCP 8.5 is consistently less than the historic discharge for all the eight Kenyan rivers for the 2015–2030 and 2031–2050 periods (Figure 5a,b).

**Table 1.** Estimated coefficient of variation of discharge for rivers flowing into Lake Victoria for 2015–2100.

| | | | % Coefficient of Variation | | | | | | | | | | | |
|---|---|---|---|---|---|---|---|---|---|---|---|---|---|---|
| | | | **2015–2030** | | | **2031–2050** | | | **2051–2070** | | | **2071–2100** | | |
| | **Eastern rivers** | **Catchment area (Km$^2$)** | **RCP 2.6** | **RCP 4.5** | **RCP 8.5** | **RCP 2.6** | **RCP 4.5** | **RCP 8.5** | **RCP 2.6** | **RCP 4.5** | **RCP 8.5** | **RCP 2.6** | **RCP 4.5** | **RCP 8.5** |
| 1 | Gucha migori | 6612 | 45 | 53 | 43 | 51 | 62 | 99 | 52 | 80 | 59 | 54 | 80 | 79 |
| 2 | North Awach | 760 | 31 | 33 | 19 | 37 | 41 | 58 | 34 | 42 | 38 | 39 | 58 | 43 |
| 3 | Nyando | 3517 | 46 | 101 | 154 | 119 | 168 | 267 | 117 | 136 | 110 | 115 | 260 | 140 |
| 4 | Nzoia | 15,143 | 47 | 57 | 28 | 60 | 54 | 81 | 53 | 74 | 41 | 67 | 63 | 87 |
| 5 | Sio | 1450 | 45 | 55 | 39 | 58 | 63 | 85 | 53 | 74 | 52 | 59 | 85 | 80 |
| 6 | Sondu | 3583 | 39 | 48 | 43 | 54 | 59 | 75 | 48 | 68 | 47 | 50 | 87 | 75 |
| 7 | S. Awach | 780 | 30 | 32 | 20 | 37 | 37 | 50 | 32 | 43 | 34 | 37 | 49 | 42 |
| 8 | Yala | 3351 | 29 | 39 | 30 | 40 | 54 | 71 | 43 | 56 | 48 | 47 | 75 | 57 |
| | **South and S. eastern rivers** | | | | | | | | | | | | | |
| 9 | Grumeti | 13,392 | 34 | 41 | 31 | 49 | 37 | 60 | 54 | 48 | 39 | 50 | 45 | 59 |
| 10 | Simiyu | 11,577 | 41 | 68 | 42 | 55 | 82 | 93 | 71 | 63 | 53 | 75 | 78 | 79 |
| 11 | E. Shore Streams | 6644 | 45 | 71 | 44 | 52 | 83 | 98 | 70 | 66 | 53 | 88 | 88 | 79 |
| 12 | Mara | 13,915 | 62 | 72 | 51 | 62 | 104 | 101 | 63 | 74 | 82 | 118 | 120 | 91 |
| 13 | Mbalageti | 3591 | 48 | 73 | 53 | 57 | 84 | 112 | 72 | 65 | 55 | 93 | 88 | 77 |
| 14 | Magogo Moame (S) | 5207 | 104 | 163 | 116 | 105 | 235 | 235 | 129 | 129 | 94 | 168 | 220 | 130 |
| 15 | Issanga (S) | 6812 | 88 | 132 | 157 | 99 | 203 | 123 | 127 | 156 | 91 | 159 | 204 | 218 |
| 16 | S. Shore Streams (S) | 8681 | 74 | 131 | 78 | 86 | 197 | 146 | 115 | 98 | 77 | 140 | 214 | 106 |
| 17 | Nyashishi | 1565 | 72 | 91 | 76 | 90 | 119 | 135 | 88 | 128 | 93 | 123 | 132 | 91 |
| | **West and North rivers** | | | | | | | | | | | | | |
| 18 | W. Shore Streams | 733 | 7 | 7 | 6 | 5 | 6 | 5 | 5 | 9 | 8 | 9 | 6 | 7 |
| 19 | Kagera | 59,682 | 17 | 26 | 18 | 21 | 30 | 30 | 23 | 28 | 27 | 34 | 28 | 32 |
| 20 | Biharamulo | 1928 | 22 | 29 | 18 | 22 | 36 | 32 | 26 | 32 | 25 | 37 | 37 | 30 |
| 21 | Bukora (W) | 8392 | 62 | 85 | 39 | 58 | 48 | 42 | 49 | 61 | 33 | 51 | 64 | 82 |
| 22 | Katonga | 15,244 | 20 | 31 | 20 | 33 | 26 | 40 | 30 | 43 | 30 | 61 | 35 | 47 |
| 23 | N. Shore Streams (N) | 4288 | 61 | 87 | 37 | 57 | 46 | 40 | 47 | 57 | 33 | 49 | 61 | 76 |

### 3.3.2. Rivers on the Southern and South Eastern part of LVB

The nine rivers flowing from the southern and the south eastern regions drain a wide region of the basin and thus experience contrasting climatic conditions. Flow discharge trajectories projected for 2015 to 2100 for the nine rivers show an upward trend in only one river. The discharge for Issanga river shows an upward trend during 2030 to 2100 under RCP 8.5. During this period, the projected average discharge level is higher than that for the historic period under both the RCP 2.6 and 4.5 scenarios, indicating upward level shifts. Grumeti will have the least CV in the 21st century. The projected discharge for the other eight rivers displays no systematic trend, but will have marked inter-annual variation across all scenarios up to 2100. This variation is characterized by striking quasi-cyclic oscillations with temporally varying amplitudes and phases and differ across scenarios mainly in the timing and magnitude of discharge peaks and troughs. The peaks and troughs are most marked for RCP 8.5, medium for RCPs 4.5, and least for RCP 2.6. Moreover, extended periods of low discharge are more prevalent under RCP 8.5 than under the other two scenarios, implying more persistent water scarcity under RCP 8.5.

The projected average flow rates for the 2015–2030 period are comparable under RCPs 2.6 and 4.5 and greater than those for the RCP 8.5 scenarios for all the nine rivers, but in the 2031–2050 period, discharge under RCP 4.5 will be higher than those for RCPs 2.6 and 8.5. By the 2051–2070 period, the average discharge is largely comparable across all three scenarios for the nine rivers. However, the projected average flow rate is consistently higher for RCP 8.5 than for RCPs 2.6 and 4.5, both of which have comparable rates for all nine rivers (Figure 6a,b). The projected average flow rates are comparable with the corresponding average base rates (1981–2000) for all the rivers. The average for RCP 8.5 tends to exceed the average base rate, most notably for the 2071–2100 period (Figure 6a). The Issanga river is anomalous in having a far higher average discharge under RCP 8.5 than under the other two scenarios across all the periods (Figure 6b). The variance of the expected average discharge for the vast majority of the nine rivers is generally highest under RCP 4.5, medium under RCP 8.5, and least under RCP 2.6 for the 2015–2030 period (Table 1). For the 2031–2050 period, the variance of the average discharge shows regional variation. It is generally largest under RCP 4.5, medium under RCP 8.5, and smallest under RCP 2.6 for the southern and western rivers. However, for the south eastern rivers, the variance is largest under RCP 8.5, intermediate for RCP 4.5, and smallest under RCP 2.6 (Table 1). For the 2071–2100 period, the variance of the average discharge shows less clear-cut variation across scenarios but tends to be highest under RCP 4.5, medium under RCP 2.6, and lowest under RCP 8.5 (Table 1).

The %CV for river flow increases through time and with increasing emission levels. However, the %CV (>73–235%) also varies strongly spatially and is the highest for three rivers (S. Shore streams, Magogo Moame, and Issanga) draining the southern part and intermediate (20–132%) for seven rivers (Grumeti, Simiyu, E.Shore streams, Mara, Mbalageti, and Nyashishi) draining the SE part (Table 1).

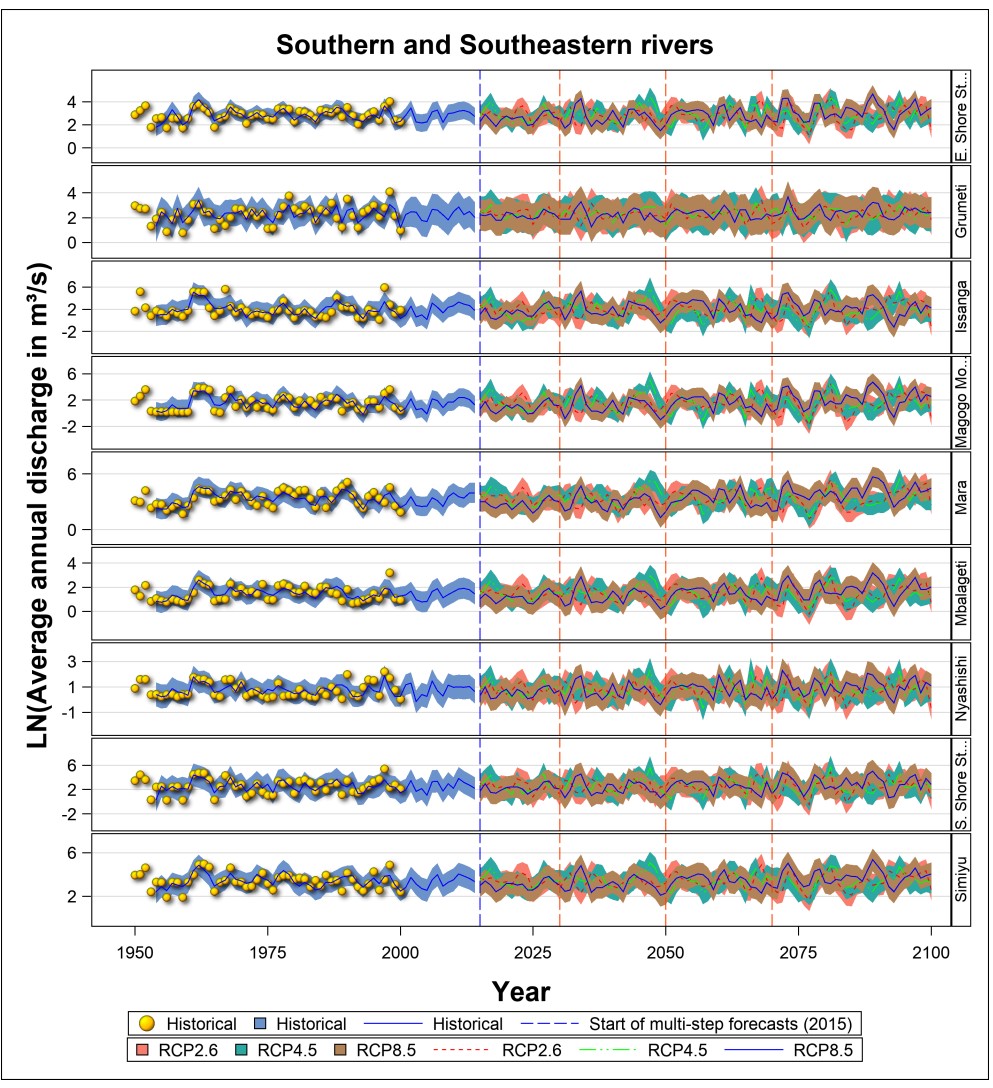

(**a**)

**Figure 6.** *Cont.*

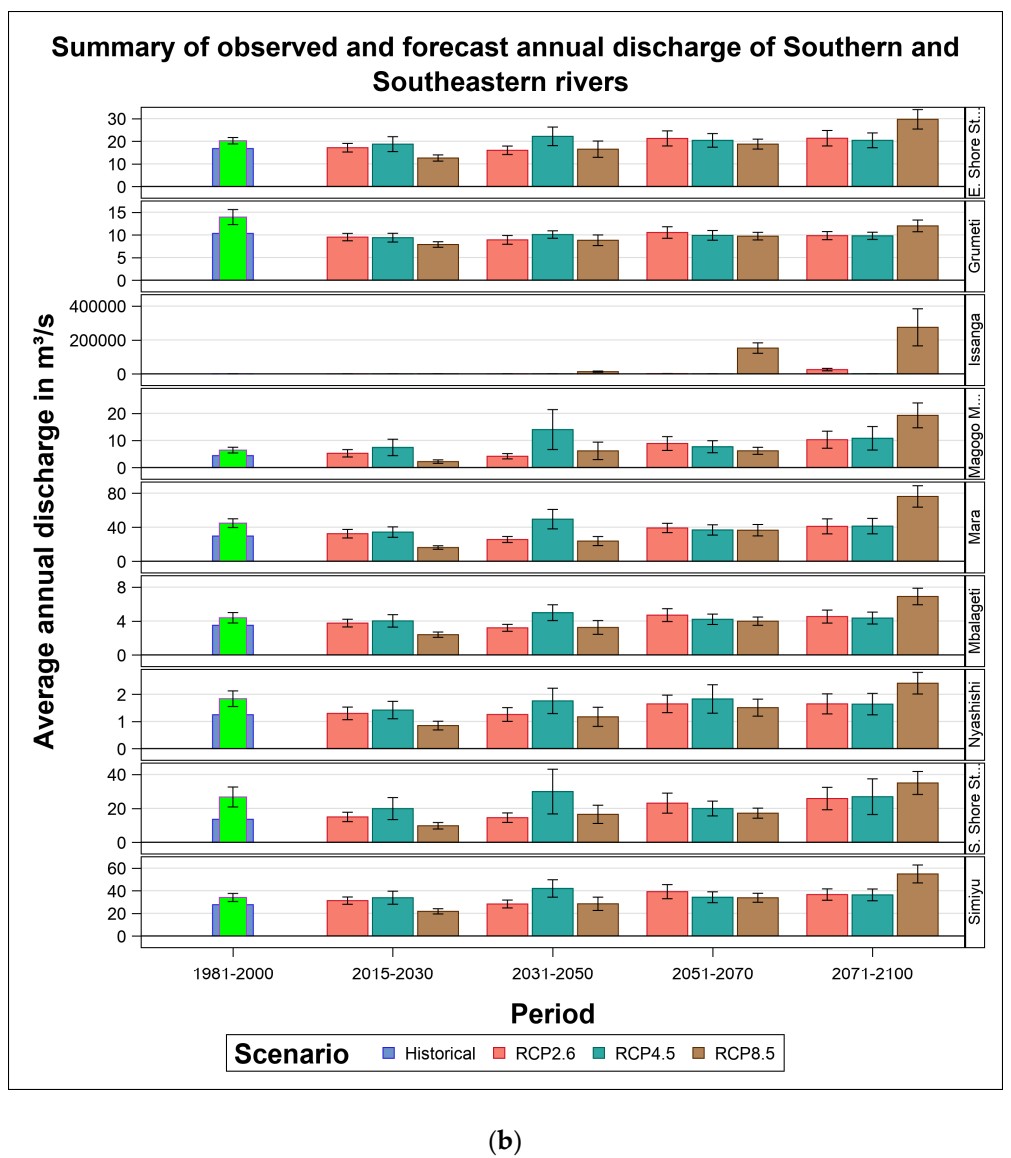

(**b**)

**Figure 6.** (**a**) Projected average annual river discharge in relation to rainfall and (**b**) summary of observed (historical) and projected average annual discharge for rivers flowing from the south and south eastern parts of the Lake Victoria Basin for the 2030s, 2050s, and 2070s under the RCP 2.6, 4.5, and 8.5 scenarios. The bright green bar is the average of the observed historic series whereas the wider medium slate blue bar is the value for the historic series predicted by the fitted model.

### 3.3.3. Rivers on the Western and Northern Part of Lake Victoria Basin

The six rivers that drain the western and northern portion of the basin will generally experience lower hydrological variability (20%), compared with the other rivers in the basin. The lowest %CV (5.3–37%) are projected for four rivers (W. Shore streams, Katonga, Kagera, and Biharamulo) during 2015–2100 (Table 1), whereas Bukora and N. Shore streams will have slightly higher variability (up to 87%), comparable to some rivers in the eastern part of the basin. The projected flow rates display no systematic trend for these rivers. While the projected average discharge for the largest river in the basin (Kagera river) is initially similar to the base period in the 2015–2070 period, it increases after 2071–2100 under RCP 8.5. The Western Shore streams will experience very little inter-annual variability under RCP 8.5 (Figure 7a).

In general, all the rivers will have marked inter-annual variation in the amplitudes of their oscillations under all three scenarios greater than the annual variation for the base period. The peaks

and troughs are most pronounced for RCP 8.5, medium for RCP 4.5, and least for RCP 2.6. Slight departures from this general pattern are apparent in the range of variability and timing for each period (Figure 7a). Moreover, all the projected average flow rates are largely comparable with the corresponding average historic base rates (Figure 7b).

The projected average flow rate is generally comparable across all three scenarios for these rivers. However, the average flow rate will be somewhat higher under higher emission scenarios (RCPs 8.5 and 4.5) during the 2030s, but in the 2050s and 2070s, the average discharge reduces at higher emission scenarios. The variation in the average flow rate for all the three rivers is highest under RCP 4.5, medium under RCP 2.6, and least under RCP 8.5 for the 2015–2030 and 2051–2070 periods. The 2031–2050 period deviates from this general pattern such that flow rate has the largest, middling, and smallest variances under RCPs 2.6, 4.5, and 8.5, respectively (Table 1). The peaks may indicate periods of flooding, whereas the troughs may indicate periods of hydrological droughts.

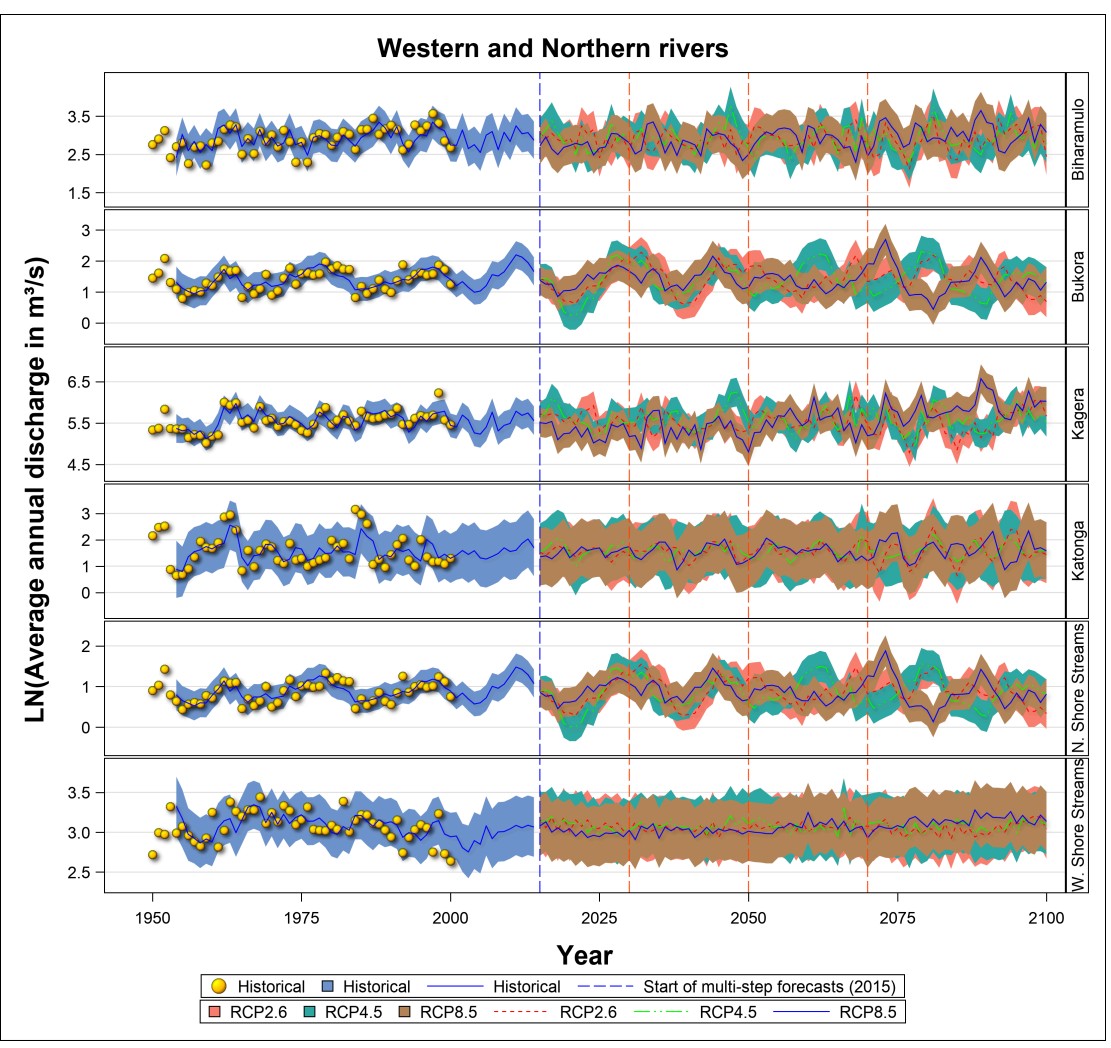

(**a**)

**Figure 7.** *Cont.*

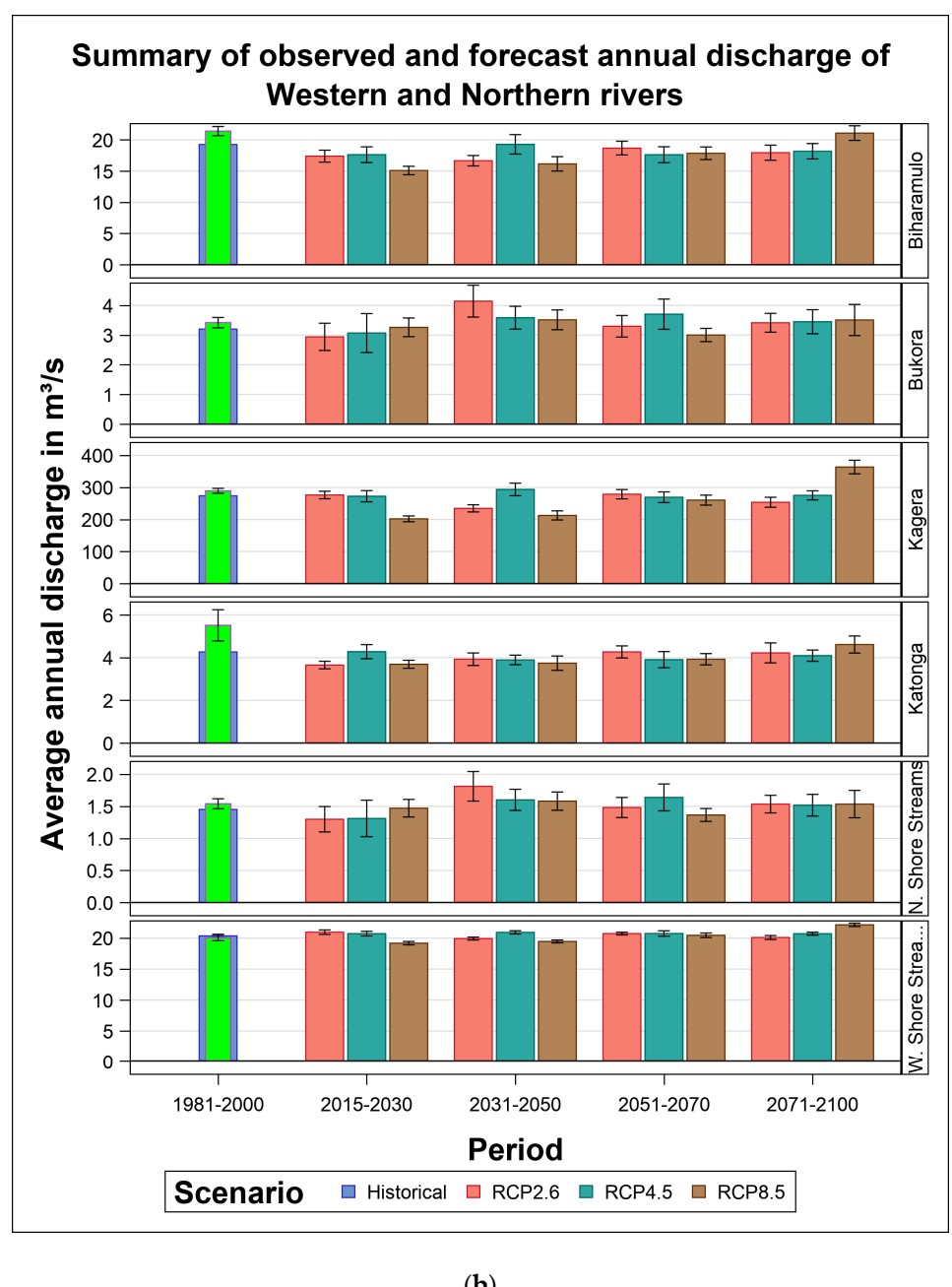

(**b**)

**Figure 7.** (**a**) Projecting average annual river discharge in relation to rainfall and (**b**) summary of the observed (historical) and projected average annual discharge for rivers flowing from the western and northern parts of the Lake Victoria Basin for the 2030s, 2050s, and 2070s under the RCP 2.6, 4.5, and 8.5 scenarios. The bright green bar is the average of the observed historic series whereas the wider medium slate blue bar is the value for the historic series predicted by the fitted model.

## 4. Discussion

### 4.1. Climate Projections

The temperature and rainfall projections for 2015–2085 exhibit significant seasonal and spatial heterogeneity over the LVB. A progressive rise in the annual minimum and maximum average temperatures of between 0.5 °C and 3.5 °C is expected throughout the basin by 2085 at higher emission scenarios (RCP 8.5) and temperature cooling is unlikely in the basin during 2015–2085. The anticipated temperature rise range is slightly lower than the 4–6 °C reported by [54] for the blue Nile area for the

2006–2085 period under RCP 8.5. The greater increase in the temperature on the southwestern side of the catchment could be linked to local processes linked to land processes. For example, soil moisture is considered a key variable in the climate system, and a negative soil moisture anomaly can lead to an increase in surface temperature through a negative anomaly of evapotranspiration [55–57]. A higher increase in minimum temperatures is projected for the cold dry season (JJAS) than for the rainy seasons (MAM and OND).

By contrast, the projections for annual rainfall show a progressive increase on the eastern and south eastern parts of the basin (parts of Kenya and Tanzania), but to decrease to below the average of the base period over the lake in the central part of the basin at higher emission scenarios. The annual distribution of this rainfall is marked by an increase over the entire basin until 2100 during the OND season. However, the MAM rainfall follows the annual pattern, and increases progressively on the eastern and south eastern portions of the basin but reduces significantly over the lake area. Though the reason for this drying out is not known, recent studies attribute it to changes in mesoscale circulation and to a secondary response to precipitation enhancement in the Congo basin [58]. The JJAS season, which is marked by minimal rainfall in parts of Uganda and Kenya, will become even drier in these regions and over the lake by up to 75% by 2085 under low (RCP 2.6) and high (RCP 8.5) emission scenarios. However, under RCP 4.5, more rainfall is expected in the southern parts of the basin. These findings contradict those of earlier studies that project an increase in the total annual precipitation of less than 10% under RCP 4.5 and less than 20% under RCP 8.5 during 2040 to 2075 [59]. However, they are consistent with the projection of a decrease in the number of rainy days in the MAM and JJAS seasons, but a slight increase in the OND season at higher emission scenarios [59]. The strong spatial variation in the projected rainfall and temperatures imply marked spatial variation in their effects on the hydrologic regimes of rivers across the basin [60,61].

The projected changes in rainfall and temperature over the basin have significant socioeconomic and ecological implications within the LVB. For example, increased rainfall expected in the south eastern parts of the LVB (Mara River catchment) during the OND season by 2051 to 2070 could directly impact forage availability for livestock and wildlife. Other major changes in the seasonal and spatial distribution of rainfall can have significant economic impacts. Moreover, the projected average temperature rise of 0.5–3.5 °C throughout the basin can adversely impact the production of crops currently grown in the LVB that cannot tolerate warmer temperatures. Further, increasing temperatures and aridity in the JJAS season can cause droughts and negatively impact the GDP of the LVB countries. Droughts are significant because they pose substantially higher impact risks to GDP than floods [15].

### 4.2. Impact of Climate Change and Variation on River Discharge and Water Resources

Annual discharges for rivers draining the LVB will be very variable in response to changes in climate and increasing emissions in the atmosphere in the coming decades. The amplitude of the oscillations in discharge will be very large after 2030 under RCP 8.5. The pronounced variability in discharge is linked to the projected increase in rainfall (10%) and temperature (by up to 3.5 °C) on the eastern parts but reduced rainfall (5% by 2070) and much higher minimum and maximum temperatures (up to 4.5° C) in the western parts of the basin. It is unlikely that the RCP 2.6 target will be achieved, thus the RCPs 4.5 and 8.5 are the most likely scenarios. Thus, lower variability (%CV) is expected for rivers on the western side of the basin: Katonga, Kagera, Western Shores, and Biharamulo. By contrast, the highest variability of over 100% is projected for the rivers in the southern portions of LVB (Magogo Moame, S. Shore streams, and Issanga).

The projected coefficient of variation (%CV) implies a high frequency of droughts and peak flows in rivers in response to changes in seasonal and annual rainfall. These would directly impact freshwater availability, base flow, stormflow, and groundwater recharge in all the river catchments within the LVB. Though, only two rivers show an increasing trend in the river hydrology, namely Nyando (RCP 4.5, East) and Issanga (RCP 8.5, south), and the %CV for 18 rivers (78%) exceeds 50%.

Thus, freshwater availability for all the towns and cities that depend on these rivers will be vulnerable to these oscillations in hydrology. Of equal importance is the hydrology of the lake, which is likely to follow the rainfall patterns because recharge to the lake is predominantly (80%) by direct rainfall whereas river inflow contributes only 20% [41].

Higher rainfall projected for the OND season is likely to increase groundwater recharge in the eastern and south eastern portions of the basin because groundwater in humid areas is sensitive to increased rainfall, though it is modified by the predominant land use/land cover [62]. Although basin scale studies are rare, regional scale studies using Gravity Recovery and Climate Experiment (GRACE) satellites have shown that the total water storage (lake and groundwater) in the LVB is positively correlated with rainfall and rainfall anomalies associated with ENSO and IOD [28,63].

With increased rainfall and discharge at higher emission scenarios, water sanitation concerns are likely to arise during flooding conditions. In many rural areas that lack good water supply and sanitation infrastructure, heavy precipitation events could increase pollution by nutrients from agricultural farms, sewer systems, and pit latrines, leading to eutrophication and turbidity of the lakes and rivers [64,65], and thus increasing water treatment costs. Another consequence of severe flooding is the destruction of water supply infrastructure with concomitant damage to property, as happened during the 1997 to 1998 record-breaking El Niño floods. Recent increases in the intensity of floods in the OND season and droughts in the MAM season in East Africa have had catastrophic consequences [1,66]. Our projections suggest that the trend of more frequent flooding will be sustained through to 2100. This trend is also consistent with the projected 10–20% increase in stream flows by 2040 [35,36].

### 4.3. Projected Impact of Climate Change and Variation on Other Economic Sectors

Given the importance of river systems and reliable rainfall to the economy of East African countries [67], we postulate the impact of the projected changes and variability on key economic sectors: Agriculture, fisheries, tourism, and energy.

Agriculture is the dominant economic activity, supporting over 80% of the population and contributes 30–40% to the gross domestic product of the basin states [68]. Farming is mainly rain-fed and increases in OND rainfall projected for the coming decades are likely to benefit the sector. However, this could mean a shift from the main farming season in MAM, whose rainfall will decline. On the contrary, projected temperature increases are likely to modify the agroclimatic zones. Thus, the development of effective early warning and water storage systems, and new more drought-resistant crop varieties would be needed to adequately deal with these anticipated changes.

The LVB is a biodiversity hotspot and changes in climate and hydrology could adversely impact its biodiversity. Climate change is shifting species distribution, composition, and phenology. The viability of populations of the remaining species and overall biodiversity will most strongly depend upon the conservation of their habitats and the prevention of pollution of the water bodies [69], which intensifies under flooding conditions. Further investigation is thus required to definitively evaluate the impact of the 4.5 °C temperature warming on the flora and fauna expected by 2070. The largest remaining migration on earth involving about 2.5 million wildebeest (*Connochaetes taurinus*), zebra (*Equus quagga burcheli*), and Thomson's gazelle (*Gazella thomsoni*) in the Mara-Serengeti and the population of mountain gorillas (*Gorilla beringei beringei*) in the headwaters of Kagera river, which are prime draw cards for tourists, depend on timely MAM and OND rainfall [70,71]. The projected changes in the amount of rainfall during the MAM season could potentially alter the ungulate migration patterns, population demography, and their dynamics [70,72–74].

Hydroelectric power (HEP) production in Sondu Miriu, Kagera rivers, and at the outlet of the Nile in Uganda and proposed on Gucha, Yala, and Nzoia rivers [75] will be impacted by the high variability in discharge projected during the 2031–2050 period, combined with high evaporation over the lake, and could reduce the average electricity generation in the drought years. Previously, severe droughts led to massive power rationing and major economic losses in the region most notably in 1999 to 2000.

While pulses of high river flow might affect some sectors positively, infrastructure and health might be negatively affected, as was the case during the 1997–1998 El Niño floods [76]. This calls for the formulation and implementation of national and regional policies and other legal instruments for effectively addressing adaptation to, and mitigation of, the effects of both floods and droughts within the river catchments.

## 5. Conclusions

To our knowledge, this is one of the first studies of its kind to explore the future (2015 to 2100) impact of climate change on the discharge of all (23) rivers from five East African countries (Burundi, Kenya, Rwanda, Uganda, and Tanzania) that flow into Lake Victoria under three different RCP emission scenarios. Our results show that EA countries need to develop climate change adaptation plans for all economic sectors dependent on freshwater supply to deal with the projected changed conditions. The changes include the following: Annual rainfall will increase throughout the catchment but with higher changes in the east than in the west; and seasonally, the MAM rainfall will reduce up to 2085 under the higher emission scenarios (RCPs 4.5 and 8.5); the projected mean annual maximum temperature will increase by 0.96–1.25 °C in the 2030s, 1.29–2.35 °C in the 2050s, and 1.19–3.6 °C in the 2070s, whereas the projected mean annual minimum temperature will increase by 1.26–1.90 °C in the 2030s, 1.5–3 °C in the 2050s, and 1.4–4.5 °C in the 2070s. The minimum temperature increase will be the greatest for the JJAS season, intermediate for the MAM season, and the least for the OND season through to the 2070s.

The VARMAX approach is suitable for projecting streamflow for large areas with transboundary basins that have inconsistencies in data availability, but have streamflow data. The VARMAX model established a strong positive linear relationship between rainfall and river discharge in the base period (1971–2000). Despite the changes in rainfall, only two rivers (Nyando and Issanga) draining the eastern and south eastern portions of the basin are projected to increase up to 2100 under the higher emission scenarios (RCPs 4.5 and 8.5). However, flow trajectories for 21 of the 23 rivers show no evident systematic trends or level shifts but pronounced oscillations with time-varying amplitudes and phases suggestive of recurrent, severe, and protracted droughts and floods at both low and high emission scenarios. The river flows are positively correlated with the future rainfall, thus the majority of the rivers in the Lake Victoria Basin will experience intermediate to high variation in flow rates. Rivers on the south eastern part of the basin (S. shore streams, Magogo Moame, and Issanga in Tanzania and Nyando in Kenya) will have the highest coefficient of variation (>50–267%) and 17 rivers draining the south eastern and eastern parts of the basin are projected to have intermediate coefficients of variation (20–120%). The least variation is expected for the rivers draining the western part of the basin (Kagera and W. shore streams), 5–33%, in response to the expected reduction in rainfall in those parts by 2100. In general, variance increases with increasing emission level across scenarios.

Climate adaptation strategies and policies should focus strongly on managing sustained high variability in river flow rates and coping with frequent severe droughts, floods, and pollution of waterways during floods. Moreover, proper management of water supply in the region is necessary to meet the growing and competing demands of the domestic, agricultural, hydropower production, industrial, wildlife conservation, and other sectors. Fresh water demand in the region will increase, driven by population growth, affluence, industrial growth, and irrigation. The periods of increased annual and seasonal (OND) runoff may produce benefits for a variety of both in-stream and out-of-stream water users by increasing available water resources, but may simultaneously increase flood risk and pollution in low-lying areas. Consequently, we recommend that proper infrastructure for waste management should be developed to enhance sanitation and reduce river pollution. Competition for water could peak during drought years, resulting in conflicts between different water users in the river catchments unless carefully managed. Thus, water allocation and rights should be carefully managed to minimize conflicts between communities, basins, economic sectors, and states. The LVB communities and states should also be well prepared to deal with disease

outbreaks, food insecurity, human–wildlife conflicts, livestock incursions into protected areas, and hydropower shortages associated with the projected recurrent droughts and floods.

**Supplementary Materials:** The following are available online at http://www.mdpi.com/2073-4441/11/7/1449/s1, Figure S1: Relationships between historical discharge and rainfall for the Lake Victoria Basin rivers in Kenya. Figure S2: Relationships between historical discharge and rainfall for the Lake Victoria Basin rivers in Uganda. Figure S3: Relationships between historical discharge and rainfall for Lake Victoria Basin rivers in Tanzania. Figure S4: Projected changes in the annual, March–April–May (MAM), June–July–August–September (JJAS), and October–November–December (OND) maximum temperature components over the Lake Victoria Basin by the 2030s. Each row corresponds to emission scenarios of RCP 2.6, RCP 4.5, and RCP 8.5. Figure S5: Projected changes in the annual, March–April–May (MAM), June–July–August–September (JJAS), and October–November–December (OND) minimum temperature components over the Lake Victoria Basin by the 2030s. Each row corresponds to emission scenarios of RCP 2.6, RCP 4.5, and RCP 8.5. Figure S6: Projected changes in the annual, March–April–May (MAM), June–July–August–September (JJAS), and October–November–December (OND) maximum temperature components over the Lake Victoria Basin by the 2050s. Each row corresponds to emission scenarios of RCP 2.6, RCP 4.5, and RCP 8.5. Figure: S7: Projected changes in the annual, March–April–May (MAM), June–July–August–September (JJAS), and October–November–December (OND) minimum temperature components over the Lake Victoria Basin by the 2050s. Each row corresponds to emission scenarios of RCP 2.6, RCP 4.5, and RCP 8.5. Figure S8: Projected changes in the annual, March–April–May (MAM), June–July–August–September (JJAS), and October–November–December (OND) maximum temperature components over the Lake Victoria Basin by the 2070s. Each row corresponds to emission scenarios of RCP 2.6, RCP 4.5, and RCP 8.5. Figure S9: Projected changes in the in annual, March–April–May (MAM), June–July–August–September (JJAS), and October–November–December (OND) minimum temperature components over the Lake Victoria Basin by the 2070s. Each row corresponds to emission scenarios of RCP 2.6, RCP 4.5, and RCP 8.5. Table S1: Estimated parameters, their standard errors, and *t*-tests of whether the parameters are significantly different from zero for the corrected Akaike information criterion (AICc)-selected best models relating river flow and annual rainfall. Table S2: Projected mean maximum temperature changes for Lake Victoria Basin for the 2030s, 2050s, and 2070s under RCPs 2.6, 4.5, and 8.5. Table S3: Projected mean minimum temperature changes for Lake Victoria Basin for the 2030s, 2050s, and 2070s for RCPs 2.6, 4.5, and 8.5. Table S4: Selection of the lagged or moving average rainfall component most strongly correlated with river discharge using the corrected Akaike information criterion (AICc). Twelve rainfall components are considered for each river. Table S5: Roots of the autoregressive (AR) and moving average (MA) characteristic polynomials for the univariate vector autoregressive moving average processes (VARMAX) model for the 23 rivers draining into Lake Victoria during 1950–2000. Table S6: Univariate model white noise diagnostics for the 23 rivers draining into Lake Victoria during 1950 to 2000, testing whether the residuals are correlated and heteroscedastic. The Durbin–Watson test statistics test the null hypothesis that the residuals are uncorrelated. The Jarque–Bera normality test tests the null hypothesis that the residuals are normally distributed. The F statistics and their *p*-values for ARCH (1) disturbances test the null hypothesis that the residuals have equal covariances. Table S7: Univariate autoregressive (AR) model diagnostics for the 23 rivers draining into Lake Victoria during 1950–2000. The F statistics and their *p*-values for the AR(1), AR(1,2), AR(1,2,3), and AR(1,2,3,4) models of residuals test the null hypothesis that the residuals are uncorrelated. Table S8: Portmanteau test for cross correlations of residuals from the univariate VARMAX (*p,q*,0) model for the 23 rivers during 1950–2000. Insignificant tests for white noise residuals based on the cross correlations of the residuals mean that we cannot reject the null hypothesis that the residuals are uncorrelated. Table S9: Univariate model ANOVA diagnostics for the 23 rivers draining into Lake Victoria during 1950–2000 show that each model is significant. Table S10: Model parameter estimates for the univariate VARMAX (*p,q,s*) model for river flow for 23 rivers draining into the Lake Victoria during 1950–2000. Table S11: Pearson correlation between components of annual rainfall and discharge of 23 rivers draining into the Lake Victoria Basin during 1950–2000.

**Author Contributions:** Conceptualization, L.A.O. and J.O.O.; methodology, J.O.O. and M.Y.S; validation, M.Y.S., J.O.O. and C.O.; writing—original draft preparation, L.A.O.; writing—review and editing, L.A.O and J.O.O.; funding acquisition, J.O.O.

**Funding:** J.O.O. was supported by a grant from the German National Research Foundation (Grant No. OG 83/1-1). This project has received funding from the European Union's Horizon 2020 research and innovation programme under Grant Agreement No. 641918. This study was also supported by The Planning for Resilience in East Africa through Policy, Adaptation, Research, and Economic Development (USAID PREPARED) Project. The Stellenbosch Institute for Advanced Study (STIAS) provided a STIAS residence fellowship to L.A.O. where much of this MS was developed.

**Acknowledgments:** We thank three anonymous reviewers for comments that helped greatly improve earlier drafts of this paper.

**Conflicts of Interest:** The authors declare no conflict of interest. The funders had no role in the design of the study; in the collection, analyses, or interpretation of data; in the writing of the manuscript, or in the decision to publish the results.

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
