# Peer review of "Projected Climatic and Hydrologic Changes to Lake Victoria Basin Rivers under Three RCP Emission Scenarios for 2015–2100 and Impacts on the Water Sector"

_water, doi:10.3390/w11071449_

Reviewer 1 Report

Comments and suggestions are in the attached file

Author Response

REVIEWER 1

The paper analyses precipitation, temperature and annual discharge behaviour for Lake Victoria basins under three RCP scenarios considering a specific modelling configuration. The rainfall-runoff transformation process is simulated using Vector Autoregressive Moving Average Process (VARMAX) calibrated on observed precipitation (CHIRPS) and temperature.

Major comments

1)    The paper is long and not easy to read, especially the descriptions of the climatic projection (line between 285 and 392) should be summarized and presented more effectively.

We have summarized the climatic projections and re-written the text in lines 285 to 392.

2)    All the analyses are based on one climatological model configuration (a specific RCM nested in a specific GCM). It is well known that the projections provided by climatic model are subject to uncertainties. The uncertainties are most challenging for extremes, for projections at local scales, and for the influence of natural climate variability. The uncertainty assessment is usually based on the results of an ensemble multiple models. If such an ensemble is not available this should be properly stated and addressed, it is the main, serious, limitation for the analysis. If ensemble is available, or its analysis has been published, the position of the chosen model configuration, with respect to the average trends of the ensemble, must be evaluated.

We agree that uncertainties are a generic feature of all statistical models. In the field of climate projections ensemble multiple models are often used to address this. We did not use ensemble multiple models but provide 95% pointwise confidence bands and projection standard errors to quantify uncertainty in the projections. Similarly, we provide standard errors of the model parameter estimates, which also measure uncertainty.

The VARMAX model can also be implemented using the Bayesian Monte Carlo Iterations. This allows simulation of many possible future discharge trajectories and quantification of uncertainties in parameter estimates and future projections in terms of 95% credible limits. The greater flexibility of the Bayesian implementation of the VARMAX model is made possible through repeated sampling from the posterior distribution. But, as with Bayesian techniques in general, this can be computationally very expensive.

3)    The          Technical            Report           Climate             Scenarios           for         Switzerland https:Unaturalsciences.ch/service/publications/107868-ch2018---climate-scenarios-for­ switzerland offers numerous ideas to improve the representation of results and discussions about uncertainty evaluation and bias correction of climatological variable.

4)     Is the VARMAX model appropriate to represent the rainfall-runoff behaviour of a basin? References about the use of this model in hydrology are missing.

      We cannot answer this question in general terms but our specific application suggests that the representation of discharge is intuitively appealing and reasonable.

 5)    The VARMAX model configuration, used to simulate the annual discharge under climate change is not clearly presented. The terms in 510 table are not described (e.g what is XLO_l_l? el? log_discharge(t-1) among variable does it means that discharge is used as variable to predict discharge?). Which is the configuration of the model used? Does it vary for each basin?

We have now provided additional descriptions of the parameters in the tables in question. The symbol referred simply means that discharge at time t is dependent on or is serially correlated with discharge up to p time steps earlier, at time t-p, as specified by the order (p) of the autoregressive model. Additionally, the residuals in discharge at time t is similarly correlated with the residuals up to q time steps earlier, t-q, as specified by the moving average order q. The model also includes lagged values of covariates such as rainfall up to lag s.

6)         Tables are hardly understandable -proper descriptions are not provided - explanation about the name of the variable and the contents of the columns are missing (e.g.table4 what is Mavrain2, AICc, etc.)

We have now provide more detailed descriptions of the tables. Mavrain2 is the

moving average of annual rainfall computed over the current and the preceding year.

AICc is the corrected Akaike Information Criterion, etc.

7)          Have been the model calibrated using CHIRPS rainfall?

The models were calibrated using the CHIRPS rainfall enriched with additional station records

by US Geological Society, Famine Early Warning System (FEWNET).

8)         Have been precipitation and temperature predicted by the RCM properly bias corrected? Which technique have been used? It is well known that climatic models have biases in the representation of temperature and precipitation that must be corrected before their quantitative use.

We received help from the Intergovernmental Centre for Climate Prediction with

the preparation of the predicted precipitation and temperature data as well as bias

correction under each of the RCM scenarios

9)          Paragraph 3.1- Why is presented this analysis? Was the linear model usedto simulate annual discharge under climate change?

Historic discharge was first related to historic rainfall and temperature using linear, nonlinear and semi-parametric models involving penalized cubic basis splines. But the linear models had the greatest strength of support in the data based on AICc and was therefore used for projection.

10)     If the model VARMAX reproduces annual discharge why in line 427 "mean monthly discharge" is mentioned as projected variable? In the captures of figures 5, 6, 7 mean monthly discharge is mentioned - please clarify.

This was an error. Mean monthly discharge has now been replaced throughout with mean

annual discharge. We thank the reviewer for catching this error.

11)     Figure 2, 3, 4 - boxplots (for specific areas) instead of maps will be probably more informative. The plotted variable is not indicated in the capture of the figure.

The figures are intended to show the spatial distribution of rainfall corresponding to the 23 catchments which the box plots will not capture.

12)     lssanga river: results in figure 6 seems unrealistic, increases in 2071-2100 are of the order of 10"3 respect to the historical and 2015-2030?

We have revised and re-run the model for Issanga. All the other results are unchanged.

13)     Comments and analyses about changes in seasonality of rainfall and temperature are extensively reported in all the paper but the hydrological model selected (VARMAX) does not allow to analyse the impact of these behaviour and trend on discharge. This should be properly stated in the paper. A more appropriate hydrological analysis could have been selected.

We related changes in historic discharge with annual and seasonal rainfall and temperature components. However, only the annual rainfall components had the strongest support in the data based on information criteria. Moreover, bivariate correlation analyses (reported in the manuscript) reinforce the conclusions arrived at based on the information criteria. Since we expected the apriori that variation in the annual and seasonal rainfall and temperature components affect variation in discharge we present all the results for completeness and transparency. As well, because the seasonal rainfall and temperature components may vary in opposing directions or to contrasting extents, as some indeed do, analysing their anticipated future variations under the three climate change scenarios yield important additional insights.

14)    This paper analyses the total annual discharge, the impact of climate change on flood trend and magnitude cannot be scientifically addressed by this work. Paragraphs 4.1,4.2 and 4.3 should be reformulated and incorporate in the introduction because they are mainly a review of literature results, the impacts of climate change illustrated in those paragraphs are not result directly linked to the results of the paper.

We do not claim to model the impact of climate change on flood trend and magnitude. Instead, we infer what the spikes and troughs in anticipated discharge trajectories imply for floods and droughts based on known and long-standing historical knowledge for this region.

Minor comments

1)    Provide the area of the basins.

The catchment area has been added in table 1

2)    Line 190 - clarify that LN is the logarithm

We have now clarified that LN is the natural logarithm in the revision.

3)    Line 397 - what do you mean for historic base line?

We mean the historic reference period. We have now made this clearer in the revision.

4)     Figure 5, 6,7 -the lower bar of the variance is missing

We have replaced the old figures with new ones with both lower and upper standard error bars.

5)    Paragraph 3.2.2 - too long

We agree and have re-written the paragraph and split it into smaller paragraphs in the revision.

6)    Table at page 12 - characters are not readable

We have updated this table and the characters are now all readable.

Reviewer 2 Report

comments are given in attachment

Author Response

Reviewer 2

Review comments:

Overall  the method introduced is interesting – replacing hydrological modelling  with derived statistics. It is especially relevant because the study  covers quite a large area and a variety of river basins.

More  attention should be given to the discussion how this method compares to  hydrological modelling and its validity under changing climate  conditions.

Abstract:

- I would consequently use  the word projections instead of predictions, the latter suggests more  certainty which is not realistic in the case of climate change.

We agree and have replaced prediction with projection throughout the text.

- Please explain VARMAX the first time it is used.

We have done this.

-  The text refers to 23 river basins and later to the Western part of the  catchment. I suggest always using the word basin when referring to the  whole Lake Victoria basin. And use catchments for the individual rivers.  As it is now it is confusing.

We agree and have done this.

Introduction:-

The introduction is a bit sloppy. There is repetition and not for all information the relevance to this study is clear.

We have removed the repetitions and synthesized the paragraphs in a better way.

-       The first two sentences bring exactly the same message.

We have revised this,

-       Rephrase the sentence on line 39

We have revised this

- Line 41 mentions many countries : is this worldwide or in Africa.

We have revised this to read ‘globally’

-       Line 50 what is the difference between agricultural output and productive actvitiy

For clarity, we have revised the sentence and removed the productive activity

- The first time LVB is used it should be fully spelled out.

We have done this.

-       line 65 I would use relation instead of dependence

Ammended

-       Line  89 ‘works for the situation with long term data coverage’ what is meant  here and is this the case for precipitation and discharge in the area?

We have revised this to show precipitation and discharge data

-       Line 94 to 97 these sentences are repetition as well.

The repetition has been removed

Materials and methods:

-       At line 120 Lake Victoria Basin is finally spelled out fully – but not needed anymore.

Done

-       The  CHIRPS product is used, but because this is a blended product its  performance and possibly its estimates vary over time – did this  influence the results of this assessments

-       Various  studies have been conducted to review the performances of CHIRPS and  rainfall data from metrological centres by regions and countries (Funk  et al., 2015; Dinku 2014; Katsanos, 2015). The CHIRPS data used in this  work were already evaluated in parts in numerous studies across East  Africa (e.g., Funk et al., 2015; Husak et al., 2013; Mukhopadhyay et  al., 2019; ).

­Climate projections:

-       line 152 explanation of RCPs is already done in relatively high detail in the Introduction is this needed at two locations?

We have removed the details from the introduction

2.4 Statistical analysis:

- the first line refers to forecasting – should this be climate projections?

Ammended to read projections

- Is the analysis done on both annual and monthly time-scales?

The  analysis was done on the annual time scale because only annual averages  of discharge data were available. We have revised the text to remove  any references to month. However, changes in projected rainfall, minimum  and maximum temperatures were analyzed at the monthly level and  aggregated to the seasonal level (long wet season-March-April-May (MAM),  short wet season—October-November-December (OND) and long dry  season-June-July-August-September (JJAS)).

- Is there a good reference for the VARMAX method – please add

Yes.

SAS Institute Inc. 2018. SAS/ETS® 15.1 User’s Guide. Cary, NC: SAS Institute Inc.

- Line 205 mentions some river characteristics – were discharge extremes also considered? Yes

-  Line 257: The model is only presented for annual values would this be  the same for other characteristics is it possible to extent with  formulas for other characteristics?

The  model is presented for annual values in the manuscript because we only  had access to annual averages of discharge. The model is very general  and can accommodate monthly or daily values, if such data are available.  As explained in the text, the model can accommodate various components  of variation in such fine-resolution data, such for instance as  seasonality, etc.

- The model is based on historic data –  how certain are you that it also applies under future climate change  conditions? Are the relations still valid when extremes are reached that  were not part of the historical period?

The  model is based on historical data and uses evolution equations to  project future changes in discharge like other systems for time series  forecasting with covariates. The uncertainly in the projected future  values are quantified by confidence limits around the projected values.  The model can be implemented in either frequentist or Bayesian mode  using Monte Carlo Iterations. The Vector Autoregressive Moving Average  Models with covariates (VARMAX) assume  short-range dependence but data sets displaying long-range dependence  can be readily modelled in the same framework using the Vector  Autoregressive Fractionally Integrated Moving Average Models with  Covariates (VARFIMAX). Out-of-sample  one step ad head and multistep-ahead forecasts are supported. These  allow reliable computation of forecast error and model selection  criterion.

The  forecasts are conditional on the projected future rainfall and  temperature and the three scenarios. It is also possible to incorporate  the uncertainty of parameters in the forecasts in the Bayesian context  and evaluate forecasts using tests of bias, efficiency, and accuracy.  Thus, while it is possible that extreme conditions may emerge in future  that widely deviate from those observed in the past, this was not the  case for any of the 23 rivers under any of the three emission scenarios.  

If  such extreme values lead to nonlinear transitions in the trajectory of  change, then nonlinear structural time series (state space) models may  be preferable. Such models use splines to capture nonlinearity. As  explained in the text, we considered such models but they had little  support in the data.

Results

-       The  results are presented with a lot of detail but read hard because no  details on the basin are given. For example line 280 mentions a river  with deviating results is there a physical reason for this?

Seem  like there is a physical reason. Katonga River is a unique river among  the rivers of the LVB, it connects lake George in the west and Lake  victoria in the east. It generally flows eastwards however during flood  conditions, part of it flows west, this could lead to  a weak to  negative correlation between rainfall and discharge

- Description of figure 2 to 4 last line: ‘each row corresponds to’ à ‘the rows corresponds to’

Each row corresponds to spatial patterns for the annual and seasonal components of rainfall or temperature.

-       Line 340 – ‘greater warming in the southwestern part’ is there an explanation for this?

We  speculate that this is linked to local processes, from previous studies  the explanation that seems plausible is related to soil moisture which  is a key variable in the climate system, a negative soil moisture  anomaly can lead to an increase in surface temperature through a  negative anomaly of evapotranspiration. This is has been projected for  the JAS season in East Africa (Koster et al 2006, Dosio et al 2016)

- Line 393: remove ‘the three’

Done

-  Line 396 : ‘we use the deviation of the projected river flow rate’ this  should be better explained. How are the seasonal changes established  and how is the change in daily variability incorporated.

By  this we mean that we projected discharge and compared (subtracted)  averaged values of the projected discharge for the 2030s, 2050s and  2070s with the average for the baseline or reference period. Seasonal  changes were computed only for the projected rainfall, minimum and  maximum temperatures and not for river discharge. To account for  seasonal and daily variability in discharge, the VARMAX or VARFIMAX  models would require the monthly or daily data. These were unfortunately  not available to us.

-       Line 427 and overall: explain how the results compare to the historical period

Done

- Line 444: give a physical explanation

Done

- Line 446 – 448: rephrase sentence

Done

-       line 478: Is there a biophysical explanation

It  is not very clear if there is one because a number of mechanisms such  as soil moisture, topography and local processes have been proposed by  previous workers. We have discussed these in the discussion.

- line 540 – 541: Rephrase second part of the sentence

Done

- line 588: ‘we infer that groundwater recharge will be more pronounced’ unclear what is mend and unclear how this comes.

We have revised to show that the impact of high rainfall is increased in gw recharge

-       line 633 – 637: Split into multiple sentences

Done

- Line 646: discussion on impact for hydropower – how does this relate to the projected changes?

We have revised this section to clearly state that HEP production relies on the rivers in the LVB

-  line 669: VARMAX is suitable for data poor regions – but in this  situation there were years of data available. How can the statistical  relations be established with little data?

The  statistical relationships require rich data to reliably establish. By  data poor we mean that we need much more limited types of data to make  the projections.

Reviewer 3 Report

Olaka et al. provide  a timely and sound manuscript that explores the future changes in  climate and its associated consequences on hydrologic regime in Lake  Victoria Basin. Authors provide an excellent background and description  of methods and did an extensive job exploring these impacts on 23 rivers  which is very impressive. They performed numerous statistical tests to  back their findings and applied these findings to come up with  suggestions for future mitigation and management for this basin. The  flow of manuscript is smooth and writing is strong. Although there are  some areas of Discussion (especially the first few paragraphs) which are  more rehashing of the Results section and therefore redundant. Overall,  this is useful and important contribution and I recommend publishing it  after they address changes/comments below.

Authors  used 3 RCPs while only using 1 Earth system model (MPI-ESM-LR). There  are so many studies which emphasize on using multiple AOGCMs to capture a  range of plausible projections and reduce uncertainty. I am not  implying to redo all analysis using multiple AOGCMs but at least they  should justify why they only used one and why did they chose this  specific model. Following on that, there is no mention of uncertainty in  this study! Every modeling study should accompany with uncertainty  analysis but the way that authors wrote this MS, reader gets the  impression that they are certain about all these projections. They need  to address uncertainties associated with their studies and shortcomings  in their research.

Authors  only used a statistical model which at the end links all changes in  discharge to precipitation, how about future changes in the region (e.g.  changes in vegetation, land-use)? Any comments on these?

My  other concern is about using two different datasets with different grid  size; one which is CHIRPS with 0.05 degree and the other one is 0.44  degree. How do these do work together? Also what is the temporal scale  of these datasets (daily, weekly)?

Last  but not least, inter-annual variability is important and in graphs  which show forecasting of discharge, they only look at annual time  series, how about monthly and seasonal? In my opinion and according to  their findings, monthly variations is more important that annual  average. Also in the supplementary materials, I only got supplementary  tables and no figures which they referenced in the text!

Below are some minor comments:

Line 18: add “period” after 2015-2100 and please do that for all the text.

L20-21: make the seasons similar to text; e.g. MAM

L40: add urbanization.

L48: reference?

L62: change “on” to “and”.

L65: define LVB before using it.

L68: add , after relatively.

L83: change it to spatiotemporal

L84: add “of” after “period”

L110: the under-construction dam does not contribute to 800 MW of power.

L118: add “for” after “supply”

L119: add “,” after “water stressed”.

L130: what is the resolution of time series?

L164: how about the period that covers 2100 since the last one ends on 2085.

L172: why January and February are missing in all the text?

L181: why six moving average? Justification? Why lag of 0-4 years?

L183: six lags or 5 (0-4?

L202-206: replace period with “,” when you list.

L234-241: ditto.

L244: is r2 of 0.35 significant??

So many statistical test but I did not find the results of all them in supplementary tables.

L267-272: define low, moderate, high correlation, what are the thresholds?

L400: Last column of the table should be 2071-2100, fix please.

L432: define LN (title of Y-axis)

L433: it seems to me that pattern is the same and the driver is only RCP across all rivers, discuss please.

L433: tile of graph has a typo; change “n” to in.

L435: What is the light green bar graph imposed on top of historical values? Nothing found in legend.

L450-475: no mention of seasonal variability.

L489: delet “(“ before “87%”

L524: delete “r” before “considered”.

L526: change ‘ to .

L598: change “whose” to “which its”.

L626-660: consolidate and remove redundancy with results, mostly is a rehash of the “Results”.

Author Response

Response to Reviewer

Comments and Suggestions for Authors

Olaka et al. provide  a timely and sound manuscript that explores the future changes in  climate and its associated consequences on hydrologic regime in Lake  Victoria Basin. Authors provide an excellent background and description  of methods and did an extensive job exploring these impacts on 23 rivers  which is very impressive. They performed numerous statistical tests to  back their findings and applied these findings to come up with  suggestions for future mitigation and management for this basin. The  flow of manuscript is smooth and writing is strong. Although there are  some areas of Discussion (especially the first few paragraphs) which are  more rehashing of the Results section and therefore redundant. Overall,  this is useful and important contribution and I recommend publishing it  after they address changes/comments below.

Authors  used 3 RCPs while only using 1 Earth system model (MPI-ESM-LR). There  are so many studies which emphasize on using multiple AOGCMs to capture a  range of plausible projections and reduce uncertainty. I am not  implying to redo all analysis using multiple AOGCMs but at least they  should justify why they only used one and why did they chose this  specific model.

The  projected climate data used in this study was generated from the Rossby  Center Regional Atmospheric Model (RCA4). This model was selected based  on a study by Endris et al., (2015) who evaluated a number of ensemble  models. This model provides one of the possible climate scenarios for  the region and it important to access other possible scenario.

We  agree that it is important to capture a range of plausible projection  scenarios to reduce uncertainty. We do this by considering the best case  (RCP 2.6), worst case (RCP 8.5) and one intermediate (RCP 4.5)  scenario. For each scenario, we provide 95% pointwise projection  confidence bands that further quantify the uncertainty associated with  each projection scenario. Many other scenarios can potentially be  considered between the two boundary scenarios but their projections can  be expected a priori to fall between the two boundary  scenarios. We also assess the performance of our model performance  against the historical time series as the benchmark. In all the cases,  the agreement between model projections and historic observations is  reasonable. Finally, since we are dealing with 23 rivers and performing  extensive model selection for each, considering a large ensemble of  potential future projection scenarios would make the model output too  extensive to condense into a single manuscript.

Following  on that, there is no mention of uncertainty in this study! Every  modeling study should accompany with uncertainty analysis but the way  that authors wrote this MS, reader gets the impression that they are  certain about all these projections. They need to address uncertainties  associated with their studies and shortcomings in their research.

We  explicitly consider model selection uncertainty in the relationships  between discharge and rainfall and use information theoretics to choose  between contending models of functional forms relating discharge to  rainfall and the specific component of rainfall (lags or cumulative  moving averages of rainfall). In particular, we use the corrected Akaike  Information Criterion to choose models and rainfall components.  Secondly, we provide standard errors of the model parameter estimates,  which quantify parameter estimation uncertainty. Thirdly, we provide 95%  pointwise projection confidence bands that quantify the uncertainty  associated with the projections for each scenario. These are included in  each graphical plot of the historic and projected future discharge  trajectories for each river. Lastly, we consider uncertainty in the  projected future scenarios by considering three RCP scenarios. Although  we consider and quantify several sources of uncertainty, we recognize  that there are many other sources of uncertainty in practice and that it  is exceedingly difficult to consider all of them comprehensively in any  single study.

Authors  only used a statistical model which at the end links all changes in  discharge to precipitation, how about future changes in the region (e.g.  changes in vegetation, land-use)? Any comments on these?

We  considered rainfall, minimum and maximum temperatures but model  selection identified the contribution of temperature as being  unimportant after accounting for rainfall. It would indeed be desirable  to consider the potential contributions of other factors such as changes  in vegetation, land use and so on. But we did not have good historical  time series on these other factors. But changes in these other factors  almost certainly affect river discharge largely by influencing rainfall,  surface run-off of rain water and retention of rain water in soils.

My  other concern is about using two different datasets with different grid  size; one which is CHIRPS with 0.05 degree and the other one is 0.44  degree. How do these do work together? Also what is the temporal scale  of these datasets (daily, weekly)?

The  0.44 degree projected dataset that is the best data resolution that is  currently available for East Africa (Endris et al. 2015). The CHIRPS  data sets were based on daily and aggregated to monthly totals. The same  was with the projected data. In these analysis we were interested in  the temporal changes and data were aggregated for the various  components.

Reference

Endris  Hussen Seid; Lennard Christopher; Hewitson Bruce; Dosio Alessandro;  Nikulin Grigory; Panitz Hans-Juergen (2015) Teleconnection responses in  multi GCM driven CORDEX RCMs over Eastern Africa. Springer. doi:  10.1007/s00382-015-2734-7

Last  but not least, inter-annual variability is important and in graphs  which show forecasting of discharge, they only look at annual time  series, how about monthly and seasonal? In my opinion and according to  their findings, monthly variations is more important that annual  average.

We  agree that there is substantial (monthly) seasonal variation in  rainfall and hence in discharge. It would thus have been interesting to  analyze interannual variation in seasonality in discharge. However, we  had access only to annual discharge data but to monthly rainfall and  temperature data. This permitted interannual forecasting of discharge  but precluded consideration of potential changes in seasonality in  discharge. But in our other recent analysis of temporal variation in  rainfall in the region, we have demonstrated that rainfall seasonality  is stably modulated by the movements of the Intertropical Convergence  Zone (ITCZ) and as result has varied little at least since 1914  (https://journals.plos.org/plosone/article?id=10.1371/journal.pone.0202814).

Also in the supplementary materials, I only got supplementary tables and no figures which they referenced in the text!

Below are some minor comments:

Line 18: add “period” after 2015-2100 and please do that for all the text. DONE

L20-21: make the seasons similar to text; e.g. MAM DONE

L40: add urbanization. DONE

L48: reference? – MOVED (ref 15) and inserted

L62: change “on” to “and”. _DONE

L65: define LVB before using it. - DONE

L68: add, after relatively.- NOT Correct grammatically

L83: change it to spatiotemporal –Done line 83

L84: add “of” after “period” – Not adopted to keep correct grammar

L110: the under-construction dam does not contribute to 800 MW of power.- correct 800MW is from 5 stations, as stated in the sentence

L118: add “for” after “supply” –.. supply  infrastructure is one compound term, we have added freshwater before supply to maintain meaning of sentence

L119: add “,” after “water stressed”. Done (line 123)

L130: what is the resolution of time series?

The  river discharge time series is annual. The rainfall and temperature  time series have a monthly resolution. The projected rainfall and  temperature series also have a monthly resolution.

L164: how about the period that covers 2100 since the last one ends on 2085.  The spatial climate projection are analysed upto 2085 this was to show  the spatial distribution for rainfall and temperature while the river  discharge  has been projected upto 2100 using the VARMAX model,

L172: why January and February are missing in all the text?

Our explanation is that we focused on the main rainy seasons (MAM and OND) and the main dry season (JJAS).

L181: why six moving average? Justification? Why lag of 0-5 years?

We  computed cumulative moving averages of prior annual rainfall spanning 0  to 5 years because annual rainfall shows irregular cylces with cycle  periods of about 5 years (see e.g.

Ogutu, J. O., Piepho, H. P., Dublin, H. T., Bhola, N., & Reid, R. S. (2008). El Niñosouthern oscillation, rainfall, temperature and normalized difference vegetation index fluctuations in the MaraSerengeti ecosystem. African Journal of Ecology, 46(2), 132-143.

Bartzke,  G. S., Ogutu, J. O., Mukhopadhyay, S., Mtui, D., Dublin, H. T., &  Piepho, H. P. (2018). Rainfall trends and variation in the Maasai Mara  ecosystem and their implications for animal population and biodiversity  dynamics. PloS one, 13(9), e0202814.)

We then selected the moving average best supported by the data using information theoretics.

L183: six lags or 6 (0-5)?

L202-206: replace period with “,” when you list. Done 2006-2015

L234-241: ditto. Done- 262-272

L244: is r2 of 0.35 significant?? The p-value is 0.0463, which is less than the threshold p-value of 0.05 and hence this significant as are all the other r2 values. 

So many statistical test but I did not find the results of all them in supplementary tables.  Results of most of the statistical tests are provided in supplementary  tables S1, S4-S11. Results of some of the statistical procedures used to  select the VARMAX model, such as selecting the orders of the  autogressive (AR) and moving average (MA) components of the VARMAX model  were omitted for brevity.

L267-272: define low, moderate, high correlation, what are the thresholds?

We define high correlation as (0.50-0.83) and low to moderate correlation as 0.31-0.49.

L400: Last column of the table should be 2071-2100, fix please. DONE  

L432: define LN (title of Y-axis)

LN is the Natural logarithm

L433: it seems to me that pattern is the same and the driver is only RCP across all rivers, discuss please.

There are some differences across rivers but are somewhat masked and harder to visualize because of

differences in the scale of discharge. But it is also largely true that most of the variation is across RCPs.

L433: tile of graph has a typo; change “n” to in.

Corrected.

L435: What is the light green bar graph imposed on top of historical values? Nothing found in legend.

Both  bars are for historical values. The chartreuse (bright green) bar is  the average of the observed historic series. The wider medium slate blue  bar is the value for the historic series predicted by the fitted model.  This is meant to help evaluate how well the model does with historic  data.

L450-475: no mention of seasonal variability.

Projected  seasonal variability was analyzed only for rainfall and temperature.  For river discharge we cab only infer what projected future fluctuations  in rainfall imply because our model uses annual resolution due to lack  of monthly data on discharge.

L489: delet “(“ before “87%” done on Line 526

L524: delete “r” before “considered”. DONE L562

L526: change ‘ to . Done

L598: change “whose” to “which its”. Grammar maintained

L626-660: consolidate and remove redundancy with results, mostly is a rehash of the “Results”. Done
